# DETAIL: Task DEmonsTration Attribution for Interpretable In-context Learning

**Zijian Zhou**[13]    **Xiaoqiang Lin**[1]    **Xinyi Xu**[12]    **Alok Prakash**[3]
**Daniela Rus**[34]    **Bryan Kian Hsiang Low**[1]

[1]Department of Computer Science, National University of Singapore, Singapore
[2]Institute for Infocomm Research, A*STAR, Singapore
[3]Singapore-MIT Alliance for Research and Technology Centre, Singapore
[4]CSAIL, MIT, USA
`{zhzijian,xiaoqiang.lin,xuxinyi,lowkh}@comp.nus.edu.sg`
`{zijian.zhou,alok.prakash}@smart.mit.edu`
`rus@csail.mit.edu`

## Abstract

In-context learning (ICL) allows transformer-based language models that are pre-trained on general text to quickly learn a specific task with a few "task demonstrations" without updating their parameters, significantly boosting their flexibility and generality. ICL possesses many distinct characteristics from conventional machine learning, thereby requiring new approaches to interpret this learning paradigm. Taking the viewpoint of recent works showing that transformers learn in context by formulating an internal optimizer, we propose an influence function-based attribution technique, `DETAIL`, that addresses the specific characteristics of ICL. We empirically verify the effectiveness of our approach for demonstration attribution while being computationally efficient. Leveraging the results, we then show how `DETAIL` can help improve model performance in real-world scenarios through demonstration reordering and curation. Finally, we experimentally prove the wide applicability of `DETAIL` by showing our attribution scores obtained on white-box models are transferable to black-box models in improving model performance.

## 1 Introduction

The rapid development of transformer-based language models [12, 14, 17, 71] has inspired a new in-context learning (ICL) paradigm [14], which allows a language model sufficiently pre-trained on general text to quickly adapt to specific tasks. This lightweight approach of customizing a general model for specific tasks is in contrast to fine-tuning [31, 68] that necessitates both access to the model parameters and resource-intensive step of tuning these parameters for model adaptation. In ICL, a few task demonstrations are included in the input text (i.e., prompt) together with a query to help the language model better understand how to answer the query. It has been shown that including task demonstrations in the prompt can enhance the capability of language models to apply common sense and logical reasoning [65, 72] and learn patterns from the supplied demonstrations [14], significantly enhancing the flexibility and generality of language models. In the ICL paradigm, each demonstration can be viewed as a "training data point" for ICL. Analogous to how the performance of a conventional supervised machine learning (ML) model depends on the quality of training data, the performance of ICL depends on the quality of task demonstrations [41]. A research question naturally arises: How to attribute and interpret ICL demonstrations that are helpful or harmful for model prediction?

Though there are many prior works on interpreting and attributing model prediction for conventional ML models [25, 34, 52], these methods are not readily applicable to ICL due to its unique charac-

teristics. Firstly, many existing attribution techniques require either computing the gradients [58] or multiple queries to the model [19], both of which are slow and computationally expensive. In contrast, ICL is often applied in *real-time* to a large foundation model [12] that necessitates the attribution approaches for ICL to be fast and efficient. Secondly, ICL is known to be sensitive to ordering: The same set of demonstrations can result in significantly different model performance under different permutations [42, 44]. However, conventional methods do not explicitly consider the ordering of training examples. Thirdly, ICL demonstration is usually supplied as a sentence comprising a sequence of tokens, rendering conventional token-level attribution methods ineffective, as they do not capture contextual information of each ICL demonstration [6, 58]. Lastly, ICL does not update model parameters, rendering conventional techniques that analyze model parameter change [34] not applicable. Moreover, the absence of the need to update model parameters also allows a good attribution result for ICL to be transferable across different language models.

To address these challenges, we propose `DETAIL`, a novel technique that takes advantage of a classical attribution approach while tackling the unique characteristics of ICL. We adopt the perspective that transformers formulate an internal optimizer [3, 16, 61, 69] for ICL. Based on this internal optimizer, we design a method to understand the impact of each demonstration on the transformer's prediction. Notably, this approach allows us to leverage powerful existing analysis tools for transformer-based models in ICL, where otherwise the characteristics of ICL make applying these tools difficult.

Specifically, we describe an intuitive (re-)formulation of the influence function [34], a popular attribution method for conventional ML, on the internal optimizer and show that `DETAIL` addresses the challenges of computational cost, sensitivity to order, and attribution quality. Then, we empirically verify that our formulation can identify demonstrations helpful for model prediction and outlier demonstrations. Additionally, we apply our method to tasks with real-world implications including prompt reordering, noisy demonstration detection, and demonstration curation to show its effectiveness. We demonstrate that `DETAIL` achieves improved performance when applied to typical white-box large language models (LLMs). Furthermore, as many powerful LLMs are currently closed-source (thus black-box), we show that our `DETAIL` score obtained on a white-box LLM (e.g., Vicuna-7b [78]) exhibits transferable characteristics to the performance on a popular black-box model (ChatGPT).

## 2 Related Work

**Understanding in-context learning.** Prior works have attempted to understand the ICL capability of transformer-based language models [2, 3, 7, 10, 14, 20, 24, 37, 49, 51, 61, 69, 74]. [14] empirically demonstrated that language models can act as few-shot learners for unseen NLP tasks. [49] explained ICL by viewing attention heads as implementing simple algorithms for token sequence completion. [51, 69] studied the ICL capability of transformers by casting it as an implicit Bayesian inference. [3, 10, 24] further showed that transformers can learn (regularized) linear and discrete functions in context. [7] showed that transformers can adaptively implement appropriate algorithms for ICL. [37] provided a statistical generalization bound for ICL. [20, 61, 62, 74] mathematically showed that transformers with specific parameters can perform gradient descent on parameters of an internal optimizer given the demonstrations. [2, 74] further proved that the parameters (of the internal optimizer) can converge during forward passing. Inspired by the theoretical grounding, which is the focus of these works, we design our novel attribution technique for task demonstrations by adopting a similar view (i.e. transformers learn in context by implementing an optimization algorithm internally).

**Data attribution.** Past works have focused on explaining and attributing model performance to training data of conventional ML [8, 19, 25, 26, 34, 40, 52, 56, 70, 79]. The rise of LLMs has inspired research efforts on attribution w.r.t. prompts with a focus on task demonstrations [11, 43, 45, 54, 73], which is distinct from training data attribution since demonstrations are provided in context. Specifically, [43] used human annotators to evaluate the verifiability of attributing model answers to a prompt. [11, 73] relied on LLMs to evaluate attribution errors. These prior works are either computationally heavy (requiring additional queries of LLMs) or time-consuming (requiring human annotators). [54] proposed an interpretability toolkit for sequence generation models using gradient- and perturbation-based methods. [45], a contemporary work, proposed to use a decoder module on the token embeddings for per-token attribution but requires costly training to learn the decoder weights. Moreover, these methods do not specifically target demonstration attribution. Some prior

techniques [19, 25] can be adapted for attributing ICL in LLMs but may be costly or ineffective. We empirically compare our method with those and the attribution methods consolidated in [54].

**Attribution in LLMs.** Past works have attempted to apply various methods including influence in attributing language models [26, 32, 35, 38, 39, 48, 63, 53, 67, 68, 75]. [48] considered a simplification of the influence function for task demonstration curation. [26, 35, 68] applied influence to pre-training and fine-tuning data of LLMs. [75] used influence to select demonstration inputs for annotation. [53] builds a classifier on the embeddings of demonstrations using a small LLM and computes influence w.r.t. the classifier for demonstration selection. In contrast, we demonstrate various use cases of our method including on-the-fly demonstration curation, reordering, and noisy demonstration detection. A contemporary work that shares technical similarity [53] focuses on demonstration selection whereas we focus on attribution and [53] is shown to be less effective than our method in Sec. 5.3. Additionally, compared to prior works leveraging influence to address specific problems, we apply influence function to provide a *general attribution* for demonstrations, with many applications that we empirically show.

## 3    Preliminaries

**In-context learning (ICL).** ICL is a learning paradigm that provides a few task demonstrations with formatted input and output for a pre-trained transformer (e.g., an LLM) to learn the mapping function from inputs to outputs in context (i.e., via the forward pass) [14]. Formally, a transformer model $f$ takes in a prompt $p(\mathcal{S}, x_{\text{query}})$ comprising a formatted sequence of demonstrations $\mathcal{S} = (z_1, z_2, \ldots, z_n)$ where $z_i = \{x_i, y_i\}$ from a specific downstream task along with a query input $x_{\text{query}}$ (i.e., prompt) to predict its label as $\hat{y}_{\text{query}} = f(p(\mathcal{S}, x_{\text{query}}))$. An visual for an example prompt is provided in App. C. We wish to attribute the model prediction $\hat{y}_{\text{query}}$ to each $z_i \in \text{set}(\mathcal{S})$.

**ICL as implementing an internal optimizer.** With the growing interest in the internal mechanism of transformers, previous works [3, 61, 62, 74] have theoretically shown that ICL can be treated as the transformer implementing an internal optimizer on the ICL demonstrations. Specifically, [61, 62, 74] formulated the objective of ICL optimizer with (regularized) mean-squared error on a linear weight applied to the token itself in linear self-attentions (LSAs) [74] or a transformation of tokens (i.e., kernelized mean-squared error) if an extra multi-layered perception is attached before the LSA [61, Proposition 2] in a recurrent transformer architecture. The transformer layers then function as performing gradient descent on the weight to minimize the objective [61, 74].

**Influence function.** Influence function [34] approximates the change of the loss of a test data point $z_{\text{test}}$ when up-weighting a training data point $z_i$. Formally, the influence of $z_i$ on predicting $z_{\text{test}}$ is[1]

$$\mathcal{I}(z_i, z_{\text{test}}) \coloneqq \nabla_\theta L(z_{\text{test}}, \hat{\theta})^\top \mathcal{I}_{\text{reg}}(z_i) = \nabla_\theta L(z_{\text{test}}, \hat{\theta})^\top H_{\hat{\theta}}^{-1} \nabla_\theta L(z_i, \hat{\theta}) \tag{1}$$

where $L(z_{\text{test}}, \hat{\theta})$ refers to the loss (function) on a test point $z_{\text{test}}$ of the model parameterized by $\hat{\theta}$ and $H_{\hat{\theta}} \coloneqq 1/n \sum_{i=1}^n \nabla_\theta^2 L(z_i, \hat{\theta})$ is the Hessian. However, this definition cannot be directly applied to ICL since there is no model parameter change during ICL, unlike the learning settings in [8, 34]. We show how to adapt the formulation of Eq. (1) to ICL in our proposed method DETAIL next.

## 4    Influence Function on Internal Kernel Regression

Following the idea that transformers learn in context by implementing an internal kernelized least-square objective, we present our formulation of DETAIL by computing the influence function on a kernelized linear regression [28]. Specifically, we build the regression w.r.t. the following kernel

$$k(x, x') \coloneqq m(x)^\top m(x') \tag{2}$$

where $m(x) \in \mathbb{R}^{1 \times d}$ refers to (the mapping of an ICL demonstration[2] to) an internal representation of $x$ (e.g., hidden state of a transformer layer) with output dimension $d$. Let $X \coloneqq (x_1, x_2, \cdots, x_n)$

---

[1]Following [8] and the experiment implementation in [34], we drop the negative sign in our influence definition. The interpretation is that higher values imply a more positive impact.

[2]For LLMs, each demonstration may consist of more than 1 token. We discuss how to address this in Sec. 5.2.

and $Y := (y_1, y_2, \cdots, y_n)$ be the vectors of inputs and outputs in $\mathcal{S}$ respectively. The equivalent kernel regression can be written as $\hat{Y} := m(X)\beta$ where $\beta \in \mathbb{R}^{d \times 1}$ is the weight vector over the kernelized feature space. In practice, the dimension $d$ of $m$ is usually much larger than the number of demonstrations, causing severe over-parameterization. Such over-parameterization renders the influence values fragile [9]. As such, we follow [9] and adopt an $\ell_2$ regularization on $\beta$ controlled by a hyper-parameter $\lambda$, which forms a *kernelized ridge regression* [47] with loss:

$$L(x, y) = [m(x)\beta - y]^2 + \lambda \beta^\top \beta . \tag{3}$$

Taking the 2nd derivative of Eq. (3), we obtain the hessian $H_\beta$ as

$$H_\beta := (1/n) \sum_{i=1}^n \nabla_\beta^2 L(x_i, y_i) = (2/n) \sum_{i=1}^n \left( m(x_i)^\top m(x_i) + \lambda I \right) . \tag{4}$$

Adopting a matrix multiplication form for the summation in Eq. (4), we write the influence of training data on the model parameters $\beta$ as follows,

$$\mathcal{I}_{\text{reg}}(z) := H_\beta^{-1} \nabla_\beta L(x, y) = n(K + \lambda I)^{-1}[m(x)^\top(m(x)\beta - y) + \lambda \beta] \tag{5}$$

where $K := m(X)^\top m(X) \in \mathbb{R}^{d \times d}$ is the Gram matrix and $\mathcal{I}_{\text{reg}} \in \mathbb{R}^d$ refers to the influence of a particular demonstration $(x, y)$ w.r.t. the kernel regression weights $\beta$. Then, combining Eqs. (1), (3) and (5), we can express the DETAIL score as the influence of a demonstration $z$ on a query $z_{\text{test}}$:

$$
\begin{aligned}
\mathcal{I}(z_{\text{test}}, z) &:= \nabla_\beta L(x_{\text{test}}, y_{\text{test}})^\top \mathcal{I}_{\text{reg}}(z) \\
&= n[m(x_{\text{test}})^\top(m(x_{\text{test}})\beta - y_{\text{test}}) + \lambda \beta](K + \lambda I)^{-1}[m(x)^\top(m(x)\beta - y) + \lambda \beta]
\end{aligned} \tag{6}
$$

where $\beta$ has a closed-form expression (shown in Alg. 1 in App. B). While inverting matrices in Eq. (6) requires $\mathcal{O}(d^3)$ time, $d$ is usually in the thousands: A typical LLM like Llama-2-7b [60] has an embedding size $d = 4096$, allowing reasonable computation time of $\mathcal{I}$ (e.g., a few seconds). In practice, this computation is accelerated by the techniques already implemented in existing scientific computation libraries admitting sub-cubic complexity for matrix inversion.

**Computing self-influence.** One important application of the influence function in ML is identifying outliers via self-influence [8, 34]. The conventional definition of $\mathcal{I}$ trivially admits computing self-influence simply by replacing $z_{\text{test}}$ in Eq. (1) with $z_i$. While the same approach applies to DETAIL, there are two shortcomings: (i) As the embedding is sensitive to the position, placing the same demonstration at the end of the prompt (as a query) or in the middle (as a demonstration) results in different embeddings, leading to unreasonable influence score. (ii) For each demonstration, it needs one forward pass of the model to compute the self-influence, which can be costly when the ICL dataset size is large. Instead, we implement self-influence for DETAIL by *reusing* the demonstration's embedding. This way, we keep the two sides of Eq. (1) consistent and only require one forward pass of the model to compute the self-influence for all demonstrations.

**Further speedup via random matrix projection.** While the current formulation in Eq. (6) is already computationally cheap, a relatively large embedding size (e.g. $d = 4096$ for Llama-2-7b) can become a bottleneck as inverting the matrix can be relatively slow. We apply an insight that for ICL, much of the information in the embedding $m$ is redundant (we do not need a 4096-dimensional $\beta$ to fit 20 demonstrations). Hence, we project $m$ to a much lower dimensional space via a random matrix projection while preserving the necessary information, following the Johnson-Lindenstrauss lemma [21, 33], precisely represented as a projection matrix $P \in \mathbb{R}^{d \times d'}$ with each entry i.i.d. from $\mathcal{N}(0, 1/d')$. We provide a more detailed discussion in App. C. Empirically, we show that we can compress $m$ to a much smaller dimension $d' \leq 1000$, resulting in a $10\times$ computation speedup on a typical 7B LLM on an NVIDIA L40 GPU (see Sec. 5.2).

A visualization of our proposed method is in Fig. 1 and a pseudo-code implementation is in App. B.

## 5 Empirical Evaluation

We evaluate the effectiveness of DETAIL (i.e., Eq. (6)) on two metrics: computational time (via logged system time required) and effectiveness in attribution (via performance metrics for tasks). We start by visualizing how the DETAIL scores, particularly $\mathcal{I}(z_{\text{test}}, z_i)$ (test influence, abbreviated as $\mathcal{I}_{\text{test}}$) and $\mathcal{I}(z_i, z_i)$ (self influence, abbreviated as $\mathcal{I}_{\text{self}}$) following [34, Sections 5.1 & 5.4], attribute demonstrations to a query first on a custom transformer and then on LLMs. Note that the hyper-parameter $\lambda$ varies under different scenarios and we discuss some heuristics for setting $\lambda$ in App. C.

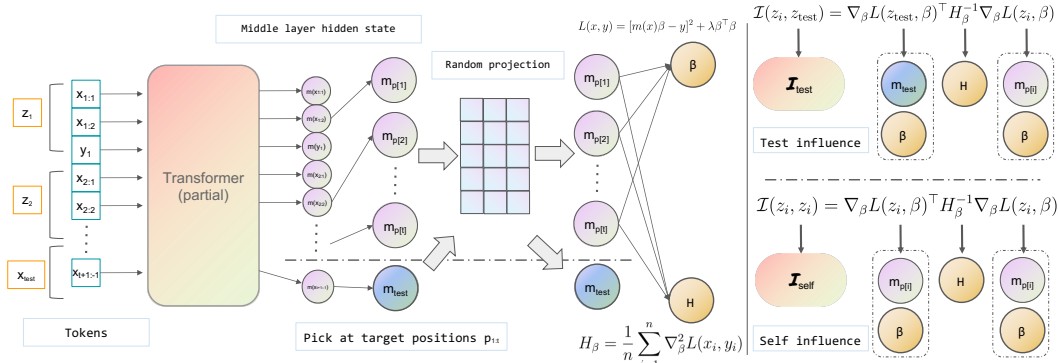

Figure 1: Illustration of computing `DETAIL` score for transformer-based ICL. Note that we use the same notation $m_{p[\cdot]}$ before and after the random projection since the projection is optional.

**Enforcing ICL behavior.** We consider tasks where transformers learn from the demonstrations and form an internal optimizer. To evaluate the effectiveness of our method, we enforce the ICL behavior by mapping the labels of the demonstrations to a token that carries no semantic meaning, This way, pre-trained transformers cannot leverage memorized knowledge to produce answers but have to learn the correct answer from the supplied demonstrations. Specifically, we map all labels to one of $\{A, B, C, D, E\}$ depending on the number of classes. More details are in each section.

## 5.1 Evaluation on a Custom Transformer

We use the MNIST dataset [22] to visualize how $\mathcal{I}_{\text{test}}$ can be applied to attribute model prediction to each demonstration. We design a task where the transformer needs to learn a mapping of the digit image to a label letter in context. Specifically, for each image of $28 \times 28$ pixels, we flatten the pixels and concatenate them with a mapped label to form a 785-dimensional token vector. For simplicity of illustration, we only use images of digits in $\{0, 1\}$. For each ICL dataset, we assign to each distinct digit a letter in $\{A, B, C, D, E\}$ randomly. We build a recurrent transformer based on the design in [61] with 10 recurrent layers each consisting of 15 attention heads. We pre-train the transformer with random label mappings from randomly selected digits so that the transformer cannot memorize the mapping but has to infer from the demonstrations. We use the hidden state after the 1st layer as $m$ to compute $\mathcal{I}_{\text{test}}$. A qualitative visualization of $\mathcal{I}_{\text{test}}$'s attribution and a quantitative plot showing how the test prediction varies by removing demonstrations with the highest (lowest) $\mathcal{I}_{\text{test}}$ are in Fig. 2. Left shows that removing tokens (represented by the image pixels and the corresponding label) with the largest $\mathcal{I}_{\text{test}}$ makes the model make wrong predictions, whereas removing tokens with the lowest $\mathcal{I}_{\text{test}}$ can retain the correct model predictions. Right shows that removing tokens with the lowest $\mathcal{I}_{\text{test}}$ results in a slower decrease in prediction accuracy than removing the highest $\mathcal{I}_{\text{test}}$, demonstrating that tokens with the highest $\mathcal{I}_{\text{test}}$'s are more helpful and *vice versa*.

## 5.2 Evaluation on Large Language Models

With the insight obtained in Sec. 5.1, we now apply `DETAIL` to full-fledged LLMs. We start with demonstration perturbation and noisy demonstration detection to demonstrate that `DETAIL` can be used to interpret the quality of a demonstration. Then, leveraging these results, we further show how we can apply the `DETAIL` scores to tasks more closely related to real-world scenarios.

A distinction between LLMs and the custom transformer used above is that demonstrations in LLMs are usually *a sequence of tokens*, whereas demonstrations in the custom transformer are single tokens representing the actual numerical values of the problems (see Fig. 2). This distinction makes it difficult to find an appropriate internal representation for each demonstration (i.e., $m$). To overcome this challenge, we draw inspiration from prior works [64, 69] which suggest that information flows from input words to output labels in ICL. As an implementation detail, we take the embedding of the *last token before the label token* (hereafter referred to as the target position) in the *middle layer* where most of the information has flown to the target token positions. We include ablation studies on using different layers' embeddings in App. D.6 and using different target positions in App. D.7.

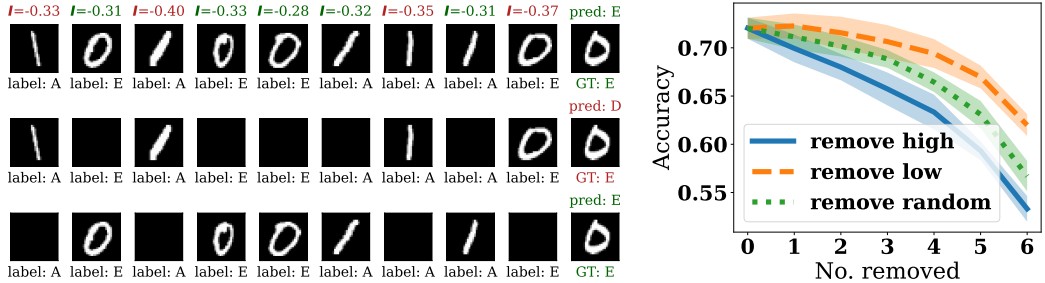

Figure 2: (Left) Visualization of learning label mapping of MNIST digits in context. The left 9 images in each row are demonstrations while the *right-most* one is a query image. Below each image shows its mapped label ("A" to "E"). Above each ICL image is its $\mathcal{I}_{\text{test}}$ w.r.t. the query image with high values highlighted in green and low values highlighted in red. Above the query image is the prediction (pred) made by the pre-trained transformer which is in green if consistent with the ground truth (GT) and red otherwise. Top row shows that using all 9 demonstrations allows the transformer to learn the mapping in context as GT=pred="E". Middle shows removing 5 demonstrations with the highest $\mathcal{I}_{\text{test}}$ results in most digit 0's removed, leading to a wrong prediction. Bottom shows removing 5 demonstrations with the lowest $\mathcal{I}_{\text{test}}$ results in 3 digit 0's remaining for the transformer to learn in context, leading to correct prediction. (Right) Average accuracy on 1013 ICL datasets repeated over 10 trials; $\lambda = 0.01$; Lines and shades represent mean and standard error over 10 independent trials.

**Setup.** We consider (for white-box models) mainly a Vicuna-7b v1.3 [78] and also a Llama-2-13b [60] on some tasks using greedy decoding to show that `DETAIL` works on models with different training data and of varying sizes. While our theoretical backing stands for transformer-based models, we experiment `DETAIL` on Mamba-2.8b [27], a state-space model architecture that has received increased attention recently in App. D.8. We primarily evaluate our method on AG News (4 classes) [77], SST-2 (2 classes) [57], Rotten Tomatoes (2 classes) [50], and Subj (2 classes) [18] datasets which all admit classification tasks. Due to space limits, some results are deferred to App. D.

**Demonstration perturbation.** We show that `DETAIL` can explain LLM predictions by showing how perturbation (i.e., corrupting the labels of some demonstrations to an incorrect class or removing some demonstrations) with the high/low $\mathcal{I}_{\text{test}}$ affects the model's predictive power. We randomly pick 20 ICL datasets each comprising 20 demonstrations and 1 query from AG News and find the average and standard error of the accuracy of predicting the query after perturbation using Vicuna-7b and Llama-2-13b, shown in Fig. 3 (results for other datasets deferred to App. D.1). It can be observed that perturbing demonstrations with low $\mathcal{I}_{\text{test}}$ results in a slower drop (or even improvement) in accuracy and *vice versa*, similar to the trend observed in Fig. 2, showcasing the applicability of `DETAIL` to LLMs. We additionally include the results using Falcon-7b [4] and Llama-2-7b [60] in App. D.1 where we perturb 10 demonstrations and observe a similar accuracy gap.

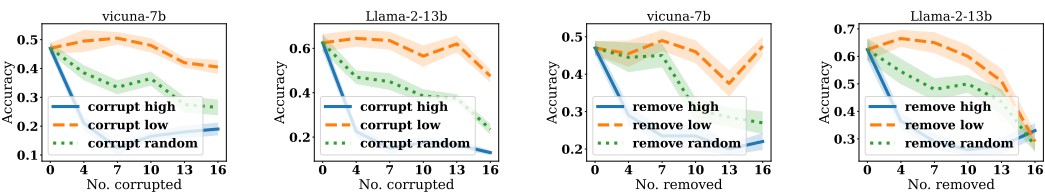

Figure 3: (1st and 2nd) Corrupting labels of demonstrations and (3rd and 4th) removing demonstrations with high/low `DETAIL` scores ($\mathcal{I}_{\text{test}}$) on AG News. Perturbing demonstrations randomly result in an accuracy in the middle as expected. All experiments are repeated 10 trials. $\lambda = 1.0$. Lines and shades represent the mean and standard error respectively.

**Noisy demonstration detection.** We utilize the `DETAIL` score to detect noisy demonstrations with corrupted labels. The experiment setup largely follows [34, Section 5.4]. We randomly draw 100 ICL datasets each consisting of 20 demonstrations and 1 query. For each ICL dataset, we randomly corrupt the labels of 4 demonstrations (i.e., flipping the label to an incorrect class). The demonstrations are

then ranked in descending order of their $\mathcal{I}_{\text{self}}$. The fraction of noisy demonstrations detected is plotted in the first 3 figures of Fig. 4 (result for other datasets deferred to App. D.2). We compare our method with the leave-one-out (LOO) score [19] where the difference in cross-entropy loss of the model output is used as the utility. It can be observed that LOO performs close to random selection, whereas our method has a much higher identification rate w.r.t. the number of demonstrations checked. We also note that our method *not only* outperforms LOO in effectiveness but is also around $10\times$ faster than LOO since LOO requires multiple LLM inferences for each demonstration in the ICL dataset.

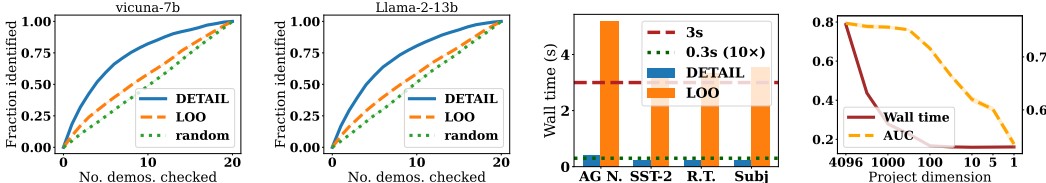

Figure 4: (1st and 2nd) Fraction of noisy labels identified vs. number of demonstrations ranked by DETAIL (with $d' = 1000$) and LOO checked on Subj using Vicuna-7b and Llama-2-13b respectively. (3rd) Wall time comparison between DETAIL and LOO on all datasets. (4th) wall time in seconds (left $y$-axis) and AUCROC (right $y$-axis) vs. projection dimension on Subj using Vicuna-7b. All experiments are repeated 10 trials. $\lambda = 10^{-9}$. Lines and shades represent the mean and std. error.

**Dimension reduction via random projection.** We analyze the impact of random projection on the effectiveness of DETAIL. Intuitively, dimension reduction trades off the effectiveness of DETAIL with computational efficiency, specifically the $\mathcal{O}(d^3)$ cost for inverting $K_{\mathcal{I}}$. To understand the trade-off, we follow the same setup as the noisy demonstration detection experiment and compare the change in AUC ROC of detection and system wall time as the dimension $d'$ of the projection matrix $P$ decreases. The result for Subj is in the last figure of Fig. 4 (results for other datasets deferred to App. D.3), showing that wall time stays minimal ($\approx 0.3$s) for project dimensions up to 1000 generally before it exponentially increases. Effectiveness measured in terms of AUC reaches optimal with $d' \geq 1000$. The results suggest a "sweet spot" – $d' \approx 1000$ – for a low running time and high performance.

## 5.3 Applications of DETAIL

With the two experiments above verifying the effectiveness of DETAIL ($\mathcal{I}_{\text{self}}$ and $\mathcal{I}_{\text{test}}$) and the experiment on random projection which ensures computational efficiency, we demonstrate next how DETAIL, with $\mathcal{I}_{\text{self}}$ for noisy demonstration detection and $\mathcal{I}_{\text{test}}$ for demonstration perturbation, can be applied to real-world scenarios, achieving superior performance and speed.

**ICL order optimization.** One distinctive trait of ICL compared to conventional ML is that the order of demonstrations affects the model's predictive performance [42, 44]. We show that $\mathcal{I}_{\text{self}}$ helps reorder the demonstrations with improved model predictive performance. We first show, using a Vicuna-7b model, that moving demonstrations with lower quality to the front (or back) of the prompt tends to improve the test accuracy of the model. To see this, we corrupt the label of a random demonstration and allocate this corrupted demonstration to different positions of the ICL dataset (each with 20 demonstrations with permutations drawn from a Sobol sequence [46] to capture the average performance better). A general trend with decreasing-then-increasing accuracy can be observed in Fig. 5: Allocating noisy demonstrations to the front (or the back) results in much higher test accuracy. Leveraging this insight, we utilize $\mathcal{I}_{\text{self}}$ to reorder a random permutation of ICL demonstrations and show the reordered prompt improves the test accuracy. For each randomly ordered prompt, $\mathcal{I}_{\text{self}}$ for each demonstration is computed (note that this computation only requires 1 pass of the LLM). Then, based on the trend observed in Fig. 5, for Subj and Rotten Tomatoes datasets, the demonstrations are reordered by placing the two demonstrations with the largest $\mathcal{I}_{\text{self}}$ in front followed by the rest in ascending order. For SST-2, the demonstrations are reordered in descending order of $\mathcal{I}_{\text{self}}$. To simulate situations where demonstrations have varying quality, we additionally consider randomly perturbing 3 demonstrations (and 6 demonstrations in App. D.4) in each ICL dataset. We note a clear improvement in test accuracy of $1.4\% \sim 3.0\%$ via reordering demonstrations *only*, as shown in Table 1. The improvement demonstrates that $\mathcal{I}_{\text{self}}$ can identify demonstrations that are low-quality or inconsistent with other demonstrations in the ICL dataset.

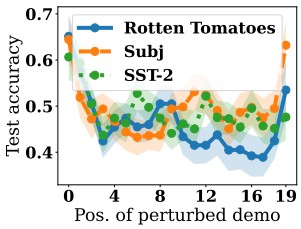

Figure 5: Test accuracy (mean and std. error) vs. position of the demonstration with the corrupted label over 50 trials.

Table 1: Predictive accuracy of demonstrations permuted randomly and based on $\mathcal{I}_{\text{self}}$, respectively. The mean and standard error (in bracket) with 80 repeated trials is shown.

|  | Subj | SST-2 | Rotten Tomatoes |
|---|---|---|---|
| **No corrupted demo** |  |  |  |
| Baseline (random) | 0.722 (7.22e-03) | 0.665 (5.24e-03) | 0.660 (1.08e-02) |
| Reorder (DETAIL) | 0.743 (7.10e-03) | 0.679 (5.42e-03) | 0.684 (1.15e-02) |
| Difference ↑ | **0.0206 (7.40e-03)** | **0.0139 (6.08e-03)** | **0.0244 (1.11e-02)** |
| **Corrupt 3 demos** |  |  |  |
| Baseline (random) | 0.655 (8.54e-03) | 0.607 (7.61e-03) | 0.553 (1.10e-02) |
| Reorder (DETAIL) | 0.685 (9.39e-03) | 0.630 (7.04e-03) | 0.582 (1.42e-02) |
| Difference ↑ | **0.0300 (9.10e-03)** | **0.0230 (7.22e-03)** | **0.0291 (1.06e-02)** |

**ICL demonstration curation.** In the demonstration perturbation experiment, we have verified that our $\mathcal{I}_{\text{test}}$ can correctly attribute helpful demonstrations w.r.t. a query. A direct application is demonstration curation where a subset of most helpful demonstrations are selected to prompt the LLM while maintaining accuracy *on a test dataset*.[3] This application is useful, especially for saving the cost of querying LLMs.[4] For proprietary LLMs, reducing the prompt length can also significantly save inference time which scales quadratically in the prompt length. As a setup, we fix a randomly selected set of 120 demonstrations as the test set. In each trial, we randomly pick 20 demonstrations to form an ICL dataset and another 20 demonstrations as the validation set. The individual $\mathcal{I}_{\text{test}}$'s on each validation demonstration are summed as the final score. Then, demonstrations with the lowest scores are removed (in position). We randomly corrupt 5 demonstrations in each ICL dataset to simulate prompts with varying qualities. The results are shown in Fig. 6 (results on other datasets deferred to App. D.5). A clear gap between the test accuracy after removing demonstrations with high/low $\mathcal{I}_{\text{test}}$ can be observed for both Vicuna-7b and Llama-2-13 on both binary (Rotten Tomatoes) and 4-way classification (AG News). Removing demonstrations with lower $\mathcal{I}_{\text{test}}$'s maintains (or even improves) the test accuracy. Moreover, the gap for the 13B model is wider and more certain (shorter error bars), signaling better curation. We attribute this phenomenon to the better capability of the larger model to formulate an "internal optimizer", which enhances the attributive power of $\mathcal{I}_{\text{test}}$.

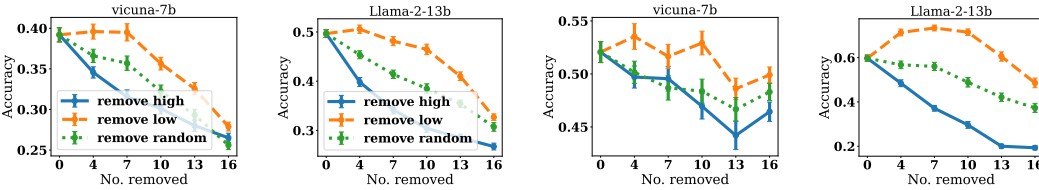

Figure 6: Test accuracy vs. number of demonstrations removed using $\mathcal{I}_{\text{test}}$ on (1st and 2nd) AG news and (3rd and 4th) Rotten Tomatoes using Vicuna-7b and Llama-2-13b. All experiments are repeated with 80 trials. Lines and bars represent the mean and standard error.

## 5.4 Comparison with Other Attribution Methods

We compare our DETAIL score with other metrics proposed for demonstration attribution/selection or can be directly extended to attributing demonstrations. We analyze both the attributability via a demonstration curation experiment and the computational cost via recording the system wall time for performing the attribution. We select representative conventional approaches from different paradigms, including integrated gradients (IG) [58] and LIME [52] (Attention [6] in App. D.5). As these methods are originally designed for token-level attribution, we use the sum of the scores of all tokens in each demonstration as the attribution score. We also compare recent efforts on demonstration selection [15, 48, 53]. In natural language processing, a popular choice for attribution has been using BERT-based scores [23, 76] to match the similarity between texts. These methods enjoy the benefit

---

[3] Note that a key difference between demonstration curation task and demonstration perturbation task is that for curation, the test dataset is unknown when computing the DETAIL scores.

[4] At the time of this writing, GPT-4 API costs $10/1mln tokens. See https://openai.com/pricing.

of fast inference since the scores are model-agnostic. However, the independence between the scores and the transformer being applied limits their interpretability and attribution accuracy. We compute the BERT score obtained on a popular sentence transformer model [5] and compare its performance against DETAIL. We select an ICL dataset of 20 demonstrations, compute the attribution scores on a validation set of 20 demonstrations, and record the accuracy after removing 10 demonstrations in place on 120 test queries. The results are tabulated in Table 2. For hyper-parameter, we choose $M = 100$ and $k = 1$ for [48], 5 iterations of LiSSA update [1] for [53], and $M = 10, K = 4$ for datamodel [15]. We do not perform batch inference for a fair comparison of computational time as some approaches do not admit batch inference. We also use a projection of $d' = 1000$ to compute $\mathcal{I}_{\text{test}}$. It can be observed that DETAIL outperforms all other attribution methods in test accuracy. Our computation is efficient (with wall time of $4 \sim 10$s), achieving over $5\times$ speedup compared to other methods except [53] and BERT score which achieve a comparable computational time to ours but a lower accuracy. Notably, IG and LIME perform close to random removal, which is likely because these methods are designed for token-level attribution and generalize poorly to demonstration-level attribution. Interestingly, we find that combing DETAIL and BERT score achieves state-of-the-art performance. We discuss the experiment setup in detail in App. D.5.

Table 2: Test accuracy after curating the ICL dataset and the incurred wall time (in seconds on one L40 GPU). The mean and std. error (in bracket) is shown with 20 repeated trials.

| Metric | DETAIL ($d' = 1000$) | IG [58] | LIME [52] | [48] | [53] | Datamodel [15] | BERT-Score [76] | Random |
|---|---|---|---|---|---|---|---|---|
| **Subj** | | | | | | | | |
| Accuracy ↑ | **0.747 (2.60e-02)** | 0.658 (2.22e-02) | 0.665 (2.41e-02) | 0.583 (2.75e-02) | 0.556 (1.38e-02) | 0.658 (2.62e-02) | 0.671 (2.43e-02) | 0.654 (2.54e-02) |
| Wall time ↓ | **5.22 (1.17e-01)** | 593 (1.20e+01) | 393 (2.44e+01) | 54.3 (3.78e-01) | 9.37 (4.19e-01) | 746 (3.42e+00) | 2.97 (3.05e-02) | N.A. |
| **SST-2** | | | | | | | | |
| Accuracy ↑ | **0.607 (2.12e-02)** | 0.458 (2.06e-02) | 0.476 (1.87e-02) | 0.513 (1.88e-02) | 0.493 (1.34e-02) | 0.477 (2.54e-02) | 0.460 (2.36e-02) | 0.469 (2.15e-02) |
| Wall time ↓ | **4.88 (1.35e-01)** | 458 (7.99e+00) | 337 (1.69e+01) | 121 (4.79e+00) | 10.6 (7.80e-01) | 713 (1.96e+00) | 2.91 (2.78e-02) | N.A. |
| **Rotten Tomatoes** | | | | | | | | |
| Accuracy ↑ | **0.555 (1.94e-02)** | 0.442 (2.13e-02) | 0.435 (1.39e-02) | 0.520 (2.17e-02) | 0.498 (1.72e-02) | 0.484 (1.87e-02) | 0.435 (1.33e-02) | 0.457 (2.19e-02) |
| Wall time ↓ | **5.11 (1.06e-01)** | 525 (1.23e+01) | 245 (6.32e+01) | 122 (4.68e+00) | 9.74 (5.57e-01) | 732 (2.10e+00) | 2.95 (4.43e-02) | N.A. |
| **AG News** | | | | | | | | |
| Accuracy ↑ | **0.412 (1.35e-02)** | 0.351 (1.65e-02) | 0.368 (1.73e-02) | 0.392 (1.42e-02) | 0.361 (1.83e-02) | 0.373 (1.31e-02) | 0.355 (1.58e-02) | 0.379 (1.70e-02) |
| Wall time ↓ | 10.4 (1.07e-01) | 1208 (2.16e+01) | 599 (1.03e+01) | 81.3 (6.05e-01) | **6.94 (4.78e-02)** | 997 (7.55e+00) | 3.15 (5.87e-02) | N.A. |

## 5.5 Transferability to Black-box Models

We evaluate the transferability of DETAIL on GPT-3.5,[6] a popular black-box model. We experiment with both the demonstration reordering and demonstration curation tasks where we compute the DETAIL scores on a Vicuna-7b model (white-box) and then test the performance on GPT. Our method produces promising results on both tasks as shown in Table 3 and Table 4 respectively. Notably, curating demonstrations with our method achieves a 17.9% average improvement in accuracy compared to random curating on the demonstration curation task (Table 4). We also note an over 2% improvement in accuracy for reordering task if we corrupt 3 demonstrations (Table 3). With no corrupted demonstration, reordering with our approach does not improve performance on GPT-3.5, which we attribute to the stronger inference power of GPT-3.5, resulting in less variance w.r.t. demonstration orders, consistent with the findings in [44].

Table 3: Accuracy (on GPT-3.5) of demonstrations (demos) permuted randomly and based on $\mathcal{I}_{\text{self}}$. Mean and std. error (in bracket) with 80 trials is shown.

| | Subj | SST-2 | Rotten Tomatoes |
|---|---|---|---|
| **No corrupted demo** | | | |
| Baseline (random) | 0.708(7.79e-04) | 0.799(7.52e-04) | 0.901(6.01e-04) |
| Reorder (DETAIL) | 0.711(7.51e-04) | 0.792(9.01e-04) | 0.909(4.84e-04) |
| Difference ↑ | **0.002(7.52e-04)** | -0.007(6.49e-04) | **0.008(6.14e-04)** |
| **Corrupt 3 demos** | | | |
| Baseline (random) | 0.628(8.21e-04) | 0.720(1.11e-03) | 0.788(1.44e-03) |
| Reorder (DETAIL) | 0.660(9.57e-04) | 0.742(1.20e-03) | 0.816(1.60e-03) |
| Difference ↑ | **0.032(8.61e-04)** | **0.022(8.92e-04)** | **0.028(1.10e-03)** |

Table 4: Accuracy (on GPT-3.5) on a test dataset of size 20 after curating 10 demonstrations from the ICL dataset. The mean and std. error (in bracket) of accuracy after removal is shown with 20 repeated trials.

| Dataset | DETAIL ($d' = 1000$) | Random |
|---|---|---|
| Subj | **0.842 (2.16e-02)** | 0.660 (3.47e-02) |
| SST-2 | **0.812 (1.96e-02)** | 0.618 (5.51e-02) |
| Rotten Tomatoes | **0.690 (4.66e-02)** | 0.420 (5.14e-02) |
| AG News | **0.515 (3.08e-02)** | 0.447 (2.73e-02) |

---

[5] https://huggingface.co/sentence-transformers/msmarco-bert-base-dot-v5
[6] We use gpt-3.5-turbo-1106. See https://platform.openai.com/docs/models/gpt-3-5-turbo.

# 6 Conclusion, Limitation, and Future Work

We tackle the problem of attributing demonstrations in ICL for transformers. Based on the well-known influence function commonly used for attributing conventional ML, we propose DETAIL, an innovative adoption of the influence function to ICL through the lens of treating the transformer as implementing an internal kernelized ridge regression. Combined with a dimension reduction technique using random projection, DETAIL can be computed in real-time with an impressive performance on various real-world related tasks such as demonstration order optimization and demonstration curation. One limitation of our approach is the need to access the internal state of the transformer, which we mitigate by additionally showing that DETAIL scores are transferable to black-box models. As a first step toward attributing demonstrations w.r.t. a transformer's internal optimizer, we hope this work serves as a building block for future research to develop attribution techniques for more generalized prompting settings such as chain-of-thought [65].

## Acknowledgments and Disclosure of Funding

This research is supported by the National Research Foundation Singapore and the Singapore Ministry of Digital Development and Innovation, National AI Group under the AI Visiting Professorship Programme (award number AIVP-2024-001). This research is supported by the National Research Foundation (NRF), Prime Minister's Office, Singapore under its Campus for Research Excellence and Technological Enterprise (CREATE) programme. The Mens, Manus, and Machina (M3S) is an interdisciplinary research group (IRG) of the Singapore MIT Alliance for Research and Technology (SMART) centre.

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

# A  Computational Resources

**Hardware.**  All our experiments about 7B white-box models are conducted on a single L40 GPU. All experiments involving 13B white-box models are conducted on a single H100 GPU. Total budget for GPT-3.5 API calls to conduct the transferability experiments in Sec. 5.5 is estimated to be around US$500 (at the rate of US$0.5/1mln tokens for input and US$1.50/1mln tokens for output).

**Software.**  All our experiments are conducted using Python3.10 on a Ubuntu 22.04.4 LTS distribution. We use Jax [13] for the experiments in Sec. 5.1 and use PyTorch 2.1.0 [5] for the experiments in Sec. 5.2 and Sec. 5.3. We adopt the implementations of the LLMs (transformer-based and SSM-based) provided in the Huggingface's "transformers" [66] system throughout this work. The NLP datasets are also obtained from Huggingface's "datasets" API [36]. The precise repository references and other dependencies can be found in the code provided in the supplemental materials.

# B  Algorithmic Implementation

We provide an implementation of computing our `DETAIL` score for LLM in Alg. 1. Line 6 shows the (optional) random projection where a random matrix $P$ is multiplied by the embeddings. In line 8, $\beta$ has a closed-form expression as the solution to a regularized ridge regression. In line 10 and line 14, we select the embeddings of the target positions. Then, depending on whether we use self-influence, we use different embeddings for computing $\nabla_\beta L$ as shown in lines 15-19. Line 20 computes the inverse hessian $H_\beta^{-1} = (K_\mathcal{I} + \lambda I)^{-1}$ before finally calculating $\mathcal{I}_i$ in line 21.

---

**Algorithm 1** `DETAIL`

---

1: **Input:**  model $M$, prompt tokens $x_{[1:n]}$, label tokens $y_{[1:t]}$, target positions $p_{[1:t]}$, total number of transformer layers $L$, transformer layer to compute `DETAIL` score $l$, regularization constant $\lambda$, projection matrix $P \in \mathbb{R}^{d \times d'}$ (default $I$)

2: $\mathcal{I}_i \leftarrow 0$ for $i \in \{1, 2, \cdots, t-1\}$

3: $h_1, h_2, \cdots, h_L \leftarrow M(x_{[1:n]})$

4: $p_{\text{demo}} \leftarrow p_{[1:t-1]}$ ▷ Remove the last target which is the test query

5: $y_{\text{demo}} \leftarrow \text{one\_hot}(y_{[1:t-1]})$ ▷ Remove the last label and convert to one-hot

6: $m_{\text{demo}} \leftarrow h_l[p_{\text{demo}}]P$ ▷ Optional dimensionality reduction

7: $K_\beta \leftarrow m_{\text{demo}} m_{\text{demo}}^\top$ ▷ $K_\beta \in \mathbb{R}^{(t-1) \times (t-1)}$ for speed-up as $t \ll d'$

8: $\beta \leftarrow [(K_\beta + \lambda I)^{-1} m_{\text{demo}}]^\top y_{\text{demo}}$

9: $p_{\text{test}} \leftarrow p_{[t-1:t]}$

10: $m_{\text{test}} \leftarrow h_l[p_{\text{test}}]$

11: $y_{\text{test}} \leftarrow \text{one\_hot}(y_{[t-1:t]})$

12: $K_\mathcal{I} \leftarrow m_{\text{demo}}^\top m_{\text{demo}}$ ▷ $K_\mathcal{I} \in \mathbb{R}^{d' \times d'}$

13: **for** $i \in \{1, 2, \cdots, t-1\}$ **do**

14: $\quad m_i \leftarrow h_l[p_{[i:i+1]}]P$

15: $\quad$ **if** self influence **then**

16: $\quad\quad \nabla_\beta L \leftarrow m_i^\top(m_i\beta - y_{\text{demo}}[i]) + \lambda\beta$

17: $\quad$ **else**

18: $\quad\quad \nabla_\beta L \leftarrow m_{\text{test}}^\top(m_{\text{test}}\beta - y_{\text{test}}) + \lambda\beta$

19: $\quad$ **end if**

20: $\quad \mathcal{I}_{\text{reg}} \leftarrow (K_\mathcal{I} + \lambda I)^{-1}[m_i^\top(m_i\beta - y_{\text{demo}}[i]) + \lambda\beta]$ ▷ Eq. (5) with the constant dropped

21: $\quad \mathcal{I}_i \leftarrow \mathcal{I}_i + (\nabla_\beta L)^\top \mathcal{I}_{\text{reg}}$ ▷ Eq. (6)

22: **end for**

23: **Return** $\mathcal{I}$

---

# C  Additional Discussion

**Potential societal impact.**  We propose an attribution technique for improving the interpretability of in-context learning. We believe our research has potential positive societal impacts in improving the safety of LLMs via filtering out corrupted/harmful demonstrations as demonstrated by our

experiments as well as saving energy by curating the demonstration, hence reducing the cost of querying LLMs. We do not find any direct negative societal impact posed by our research contribution.

**Setting $\lambda$.** Generally, there is no golden rule for the most appropriate $\lambda$ that regularizes the ridge parameters $\beta$. Intuitively, a larger $\lambda$ likely works better when the dimension of $\beta$ is large since the model tends to be over-parameterized (i.e., in a LLM). Therefore, we set a relatively large $\lambda = 1.0$ for LLMs and a relatively small $\lambda = 0.01$ for our custom transformer. When detecting noisy demonstrations, we may not want to regularize $\beta$ too much because we wish to retain the information captured by the eigenvalues of the hessian $H$ which can be eroded with a larger $\lambda$. As such, for the noisy demonstration detection task, we set a very small $\lambda = 10^{-9}$ to retain most of the information captured by $H$ while ensuring that it is invertible.

**Random projection matrix.** We recall the Johnson-Lindenstrauss (JL) lemma [21, 33].

**Theorem C.1** (Johnson-Lindenstrauss Lemma). For any $0 < \epsilon < 1$ and any integer $n$, let $d'$ be a positive integer such that

$$d' \geq \frac{24}{3\epsilon^2 - 2\epsilon^3} \log n \,,$$

then for any set $A$ of $n$ points $\in \mathbb{R}^d$, there exists a mapping $f : \mathbb{R}^d \to \mathbb{R}^{d'}$ such that for all $x_i, x_j \in A$,

$$(1 - \epsilon)\|x_i - x_j\|^2 \leq \|f(x_i) - f(x_j)\|^2 \leq (1 + \epsilon)\|x_i - x_j\|^2 \,.$$

A specific constructive proof is by setting $A := \frac{1}{\sqrt{d'}}R$ where $R_{i,j} \overset{\text{i.i.d.}}{\sim} \mathcal{N}(0,1)$.[7] In our work, we treat each embedding $m$ as $x$ and the projected embedding $mP$ as $f(x)$. The specific construction follows the abovementioned constructive proof defining

$$P := \frac{1}{\sqrt{d'}}R \sim \mathcal{N}(\mathbf{0}, \frac{1}{d'}\boldsymbol{I}) \,.$$

Empirically, our threshold $d' = 1000$ corresponds to $\epsilon \lessapprox 0.164$, ensuring a good preservation of (Euclidean) distance between points.

**ICL prompt example.** We include a visualization of a prompt for ICL below. Each input-output pair consists of a task demonstration. The query is appended at the end of the prompt with only the input and the output header.

---

**Example Prompt 1: Subj**

Input: tsai may be ploughing the same furrow once too often .
Output: B

Input: equilibrium the movie , as opposed to the manifesto , is really , really stupid
.
Output: B

\<More demonstrations...\>

Input: a friendly vacation for four old friends - two couples from college - turns ugly . . . then
Output: A

Input: he meets god and is given all the powers of god .
Output:

---

**Additional potential future directions.** Apart from applying DETAIL to more generalized prompting settings, we think it is also an interesting direction to research whether DETAIL can provide meaningful attribution for pruned [55, 59] (or distilled [29, 30]) networks. This is especially useful

---

[7]Lecture notes.

since, compared to very large models, pruned small networks can be deployed privately, admitting a straightforward application of DETAIL. Moreover, the size of the hidden states tends to be smaller, allowing for faster computation of DETAIL score.

# D  Additional Experiments

## D.1  Additional Results for Demonstration Perturbation Task

We include the full results for the demonstration perturbation task using a Vicuna-7b v1.3 model in Fig. 7 and using a Llama-2-13b model in Fig. 8. A consistent trend can be observed across different datasets using both models.

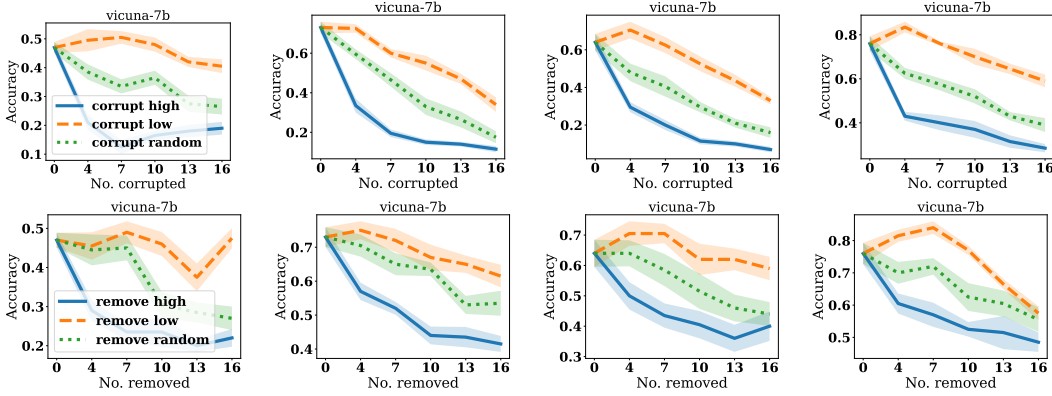

Figure 7: Corrupting and removing demonstration on datasets affects the model predictive power differently on AG News and SST-2, Rotten Tomatoes, and Subj from left to right using Vicuna-7b. Corrupting/removing demonstrations with high DETAIL scores results in lower model accuracy and *vice versa*. Corrupting/removing demonstrations randomly results in an accuracy in the middle as expected. All experiments are repeated with 10 independent trials. $\lambda = 1.0$. Lines and shades represent the mean and standard error respectively.

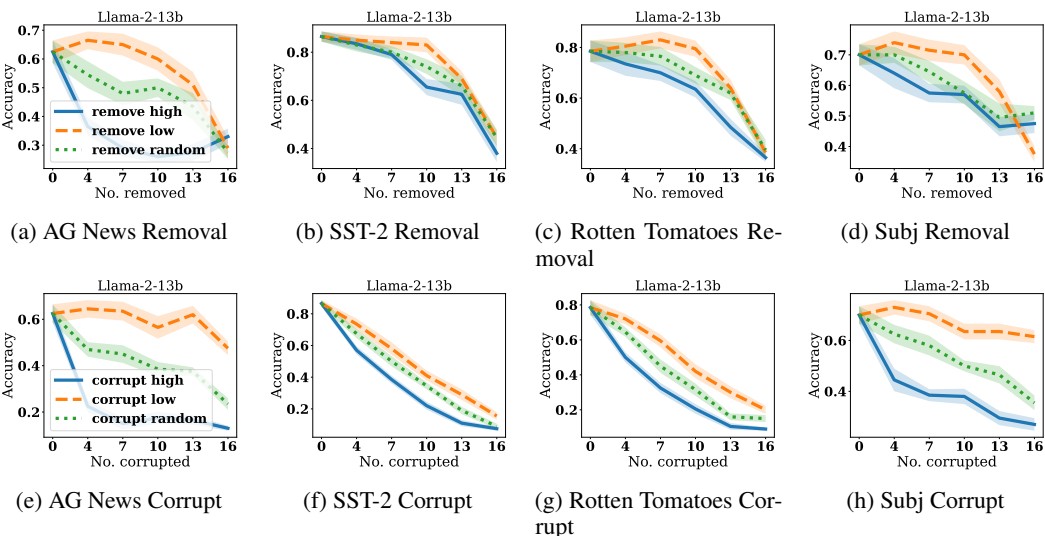

(a) AG News Removal  (b) SST-2 Removal  (c) Rotten Tomatoes Removal  (d) Subj Removal

(e) AG News Corrupt  (f) SST-2 Corrupt  (g) Rotten Tomatoes Corrupt  (h) Subj Corrupt

Figure 8: Results of model prediction accuracy vs. number of demonstrations removed/corrupted on Llama-2-13b model. $\lambda = 1.0$. Lines and shades represent the mean and standard error respectively.

We compare (in addition to Vicuna-7b) a total of 3 LLMs: Llama-2-7b [60], Llama-2-13b [60] and Falcon-7b [4]. We reuse the same experimental setup as the demonstration label perturbation task

and compare the accuracy by removing and corrupting 10 among 20 ICL data with high/low `DETAIL` scores computed in different models. The results are tabulated in Table 5. A similar trend can be observed across these models where removing/corrupting demonstrations with high $\mathcal{I}_{\text{test}}$ results in lower accuracy and *vice versa*. The results demonstrate that our method is robust against model pre-training/fine-tuning data as well as model size.

Table 5: Performance on demonstration perturbation task across different models. The mean and standard error (in bracket) of predictive accuracy after removal or corruption 10 out of 20 demonstrations of 20 randomly drawn ICL datasets is shown. All experiments are independently repeated 20 times.

| | remove high ↓ | remove low ↑ | remove random | corrupt high ↓ | corrupt low ↑ | corrupt random |
|---|---|---|---|---|---|---|
| **Subj** | | | | | | |
| Llama-2-7b | 0.380 (0.034) | 0.675 (0.025) | 0.535 (0.035) | 0.250 (0.036) | 0.690 (0.031) | 0.445 (0.019) |
| Llama-2-13b | 0.570 (0.029) | 0.700 (0.032) | 0.575 (0.038) | 0.380 (0.030) | 0.635 (0.031) | 0.500 (0.021) |
| Falcon-7b | 0.450 (0.026) | 0.690 (0.028) | 0.580 (0.042) | 0.280 (0.026) | 0.630 (0.023) | 0.445 (0.043) |
| **SST-2** | | | | | | |
| Llama-2-7b | 0.480 (0.031) | 0.670 (0.030) | 0.540 (0.042) | 0.145 (0.018) | 0.445 (0.042) | 0.275 (0.039) |
| Llama-2-13b | 0.655 (0.034) | 0.830 (0.031) | 0.740 (0.031) | 0.220 (0.025) | 0.410 (0.031) | 0.345 (0.022) |
| Falcon-7b | 0.560 (0.043) | 0.775 (0.028) | 0.680 (0.032) | 0.225 (0.031) | 0.545 (0.030) | 0.340 (0.023) |
| **Rotten toamtoes** | | | | | | |
| Llama-2-7b | 0.435 (0.034) | 0.670 (0.060) | 0.540 (0.045) | 0.120 (0.025) | 0.420 (0.039) | 0.235 (0.026) |
| Llama-2-13b | 0.635 (0.032) | 0.795 (0.032) | 0.690 (0.033) | 0.205 (0.028) | 0.420 (0.040) | 0.315 (0.035) |
| Falcon-7b | 0.475 (0.037) | 0.780 (0.024) | 0.620 (0.024) | 0.225 (0.023) | 0.590 (0.031) | 0.445 (0.025) |
| **AG News** | | | | | | |
| Llama-2-7b | 0.145 (0.018) | 0.525 (0.036) | 0.360 (0.026) | 0.150 (0.016) | 0.520 (0.030) | 0.325 (0.037) |
| Llama-2-13b | 0.260 (0.018) | 0.600 (0.041) | 0.500 (0.032) | 0.175 (0.026) | 0.565 (0.049) | 0.385 (0.031) |
| Falcon-7b | 0.155 (0.019) | 0.460 (0.021) | 0.335 (0.025) | 0.085 (0.020) | 0.465 (0.017) | 0.265 (0.027) |

## D.2 Additional Results for Noisy Demonstration Detection

We include the results on AG News, SST-2, Rotten Tomatoes, and Subj datasets using a Vicuna-7b model in Fig. 9. Similar trends as in the main text is observed. A counterpart experiment using Llama-2-13b is in Fig. 10, where a similar trend is observed.

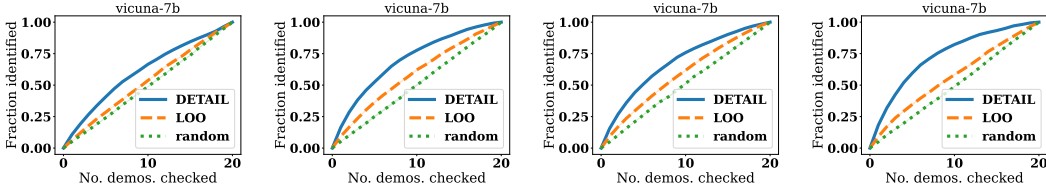

Figure 9: (Left to right) Fraction of all noisy labels identified vs. the number of demonstrations ranked by our method (with projection down to 1000 dimension) and LOO checked respectively on AG News, SST-2, Rotten Tomatoes, and Subj datasets. $\lambda = 10^{-9}$. All experiments are repeated with 10 independent trials. Lines and shades represent the mean and standard error respectively.

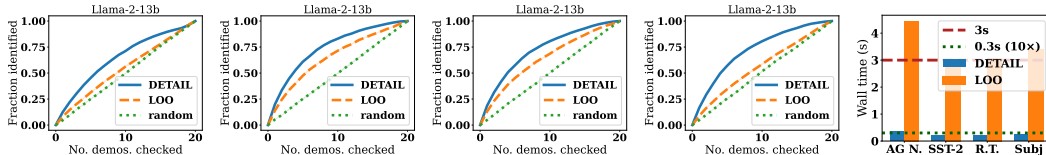

(a) Detecting noisy label on AG News (b) Detecting noisy label on SST-2 (c) Detecting noisy label on Rotten Tomatoes (d) Detecting noisy label on Subj (e) Wall time comparison

Figure 10: (a-d) Fraction of all noisy labels identified vs. the number of demonstrations ranked by our method (with projection down to $1000$ dimension) and LOO checked respectively. (e) Wall time comparison across all datasets. $\lambda = 10^{-9}$. All experiments are repeated with $10$ independent trials using a Llama-2-13b model. Lines and shades represent the mean and standard error respectively.

## D.3 Additional Results for Dimension Reduction

The experiments using Vicuna-7b on AG News, SST-2, Rotten Tomatoes, and Subj can be found in Fig. 11. It can be observed that the trend is consistent across different datasets.

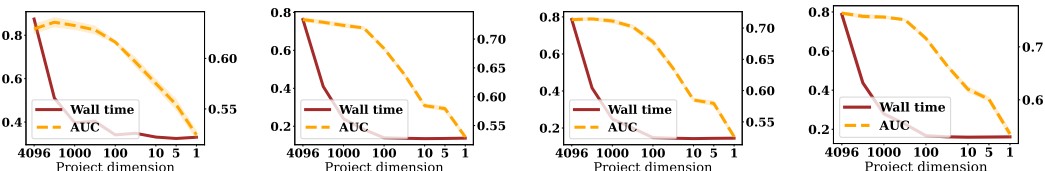

Figure 11: (Left to right) wall time in seconds (left $y$-axis) and AUCROC (right $y$-axis) vs. projection dimension $d'$ on AG news, SST-2, Rotten Tomatoes, and Subj datasets. Experiments are repeated with $10$ trials. Lines and shades represent the mean and standard error respectively.

## D.4 Additional Results for Demonstration Reordering Task

We conduct additional experiments on the demonstration reordering task by perturbing 6 demonstrations in each ICL dataset of 20 demonstrations. The results are shown in Table 6. It can be observed that reordering with $\mathcal{I}_{\text{self}}$ still achieves an improvement in test accuracy, demonstrating the robustness of our method.

Table 6: Predictive accuracy of demonstrations permuted randomly and based on $\mathcal{I}_{\text{self}}$ respectively. The mean and standard error (in bracket) with $80$ repeated trials is shown.

|  | Subj | SST-2 | Rotten Tomatoes |
|---|---|---|---|
| **Corrupt** 6 **demonstrations** | | | |
| Baseline (random) | 0.588 (7.96e-03) | 0.487 (9.45e-03) | 0.398 (1.12e-02) |
| Reorder (DETAIL) | 0.604 (7.39e-03) | 0.520 (1.01e-02) | 0.425 (1.37-02) |
| Difference ↑ | **0.0164 (7.05e-03)** | **0.0323 (8.13e-03)** | **0.0267 (1.01e-02)** |

## D.5 Additional Results for Demonstration Curation Task

We include the full results for all datasets on both Vicuna-7b and Llama-2-13b in Fig. 12. It can be observed that the gap between removing demonstrations with high/low $\mathcal{I}_{\text{test}}$ is wider with Llama-2-13b. We believe this is because Llama-2-13b being a larger model possesses better capability of formulating the internal optimizer as compared to Vicuna-7b which is smaller.

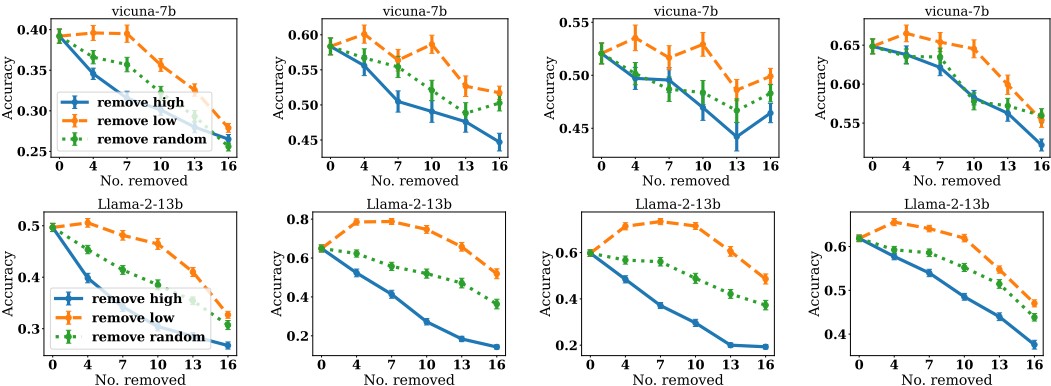

Figure 12: (Left to right) test accuracy vs. number of demonstrations removed using $\mathcal{I}_{\text{test}}$ on AG news, SST-2, Rotten Tomatoes, and Subj datasets using (top) Vicuna-7b and (bottom) Llama-2-13b. All experiments are repeated with 80 independent trials. Lines and bars represent the mean and standard error respectively.

We additionally provide the result for the demonstration curation task with Attention [6] and [53] using 10 iterations of LiSSA update for computing the hessian vector product [1] and compare it with the case with 5 iterations. The result is shown in Table 7. With 10 iterations, the wall time is much higher (over $20\times$) but accuracy is comparable to the case with 5 iterations. For Attention, the performance is comparable to random removal as shown in Table 2.

Table 7: Test accuracy after curating the ICL dataset and the incurred wall time (in seconds on one L40 GPU). The mean and std. error (in bracket) is shown with 20 repeated trials.

| Metric | Attention [6] | [53] (#5) | [53] (#10) |
|---|---|---|---|
| **Subj (curate 10 demonstrations)** | | | |
| Accuracy ↑ | 0.627 (2.01e-02) | 0.556 (1.38e-02) | 0.597 (3.23e-02) |
| Wall time ↓ | 37.6 (1.09e+00) | 9.37 (4.19e-01) | 245 (8.80e-01) |
| **SST-2 (curate 10 demonstrations)** | | | |
| Accuracy ↑ | 0.460 (2.60e-02) | 0.493 (1.34e-02) | 0.499 (1.64e-02) |
| Wall time ↓ | 24.9 (7.26e-01) | 10.6 (7.80e-01) | 241 (9.63e-01) |
| **Rotten Tomatoes (curate 10 demonstrations)** | | | |
| Accuracy ↑ | 0.488 (2.05e-02) | 0.498 (1.72e-02) | 0.494 (1.74e-02) |
| Wall time ↓ | 36.8 (1.09e+00) | 9.74 (5.57e-01) | 240 (4.61e-01) |
| **AG News (curate 10 demonstrations)** | | | |
| Accuracy ↑ | 0.350 (1.35e-02) | 0.416 (2.10e-02) | 0.346 (2.07e-02) |
| Wall time ↓ | 138 (3.28e+00) | 11.9 (5.78e-01) | 441 (4.0e-01) |

Given that BERT-score is light-weight, it is intuitive to think about whether it is possible to combine DETAIL and BERT score and reap the benefits of both worlds. We conduct a preliminary experiment in this direction. Specifically, we consider applying our formulation of DETAIL (Eq. (1)) using a hidden state modified by the BERT embedding. Specifically, we consider a weighted average of the BERT embedding and the transformer's hidden state as well as a direct concatenation of the two embeddings. The results are shown in Table 8. Surprisingly, we discover that using a weighted average of the transformer's hidden state and the BERT embedding improves the attribution accuracy.

Table 8: Test accuracy after curating the ICL dataset and the incurred wall time (in seconds on one L40 GPU). The mean and std. error (in bracket) is shown with 20 repeated trials.

| Metric | BERT Embedding Only | Weighted Average (equal weight) | Concatenation |
|---|---|---|---|
| **Subj (curate 10 demonstrations)** | | | |
| Accuracy ↑ | 0.665 (2.36e-02) | **0.758 (1.97e-02)** | 0.719 (2.75e-02) |
| **SST-2 (curate 10 demonstrations)** | | | |
| Accuracy ↑ | 0.475 (1.31e-02) | 0.579 (2.34e-02) | **0.596 (1.96e-02)** |
| **Rotten Tomatoes (curate 10 demonstrations)** | | | |
| Accuracy ↑ | 0.510 (2.10e-02) | **0.575 (2.46e-02)** | 0.570 (2.75e-02) |
| **AG News (curate 10 demonstrations)** | | | |
| Accuracy ↑ | 0.408 (2.10e-02) | **0.425 (1.76e-02)** | 0.412 (1.50e-02) |

## D.6 Ablation of Different Transformer Layers for Computing `DETAIL` scores.

We experiment with the difference in the effectiveness `DETAIL` using the embeddings of different layers. We conduct experiments on demonstration removal, demonstration perturbation, and noisy label detection tasks. The results are shown in Fig. 13. It can be observed that obtaining the `DETAIL` scores from the later layers of the model consistently produces desirable results.

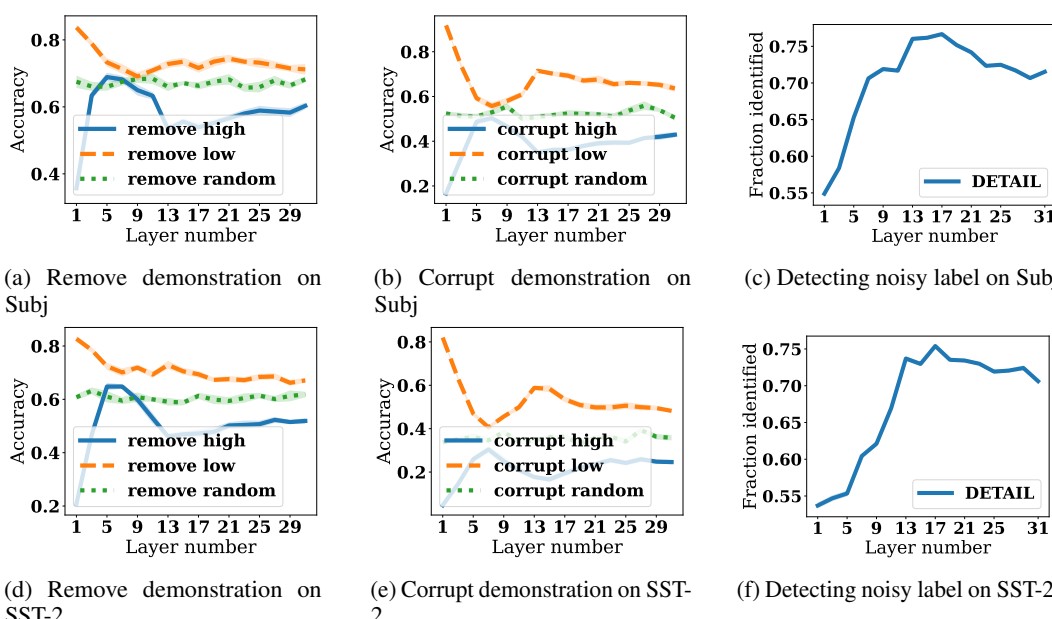

(a) Remove demonstration on Subj

(b) Corrupt demonstration on Subj

(c) Detecting noisy label on Subj

(d) Remove demonstration on SST-2

(e) Corrupt demonstration on SST-2

(f) Detecting noisy label on SST-2

Figure 13: Results of different task performance vs. the layer number in a Vicuna-7b model which consists of 31 layers. Experiments are repeated with 10 trials. $\lambda = 1.0$ for (a,b,d,e) and $\lambda = 10^{-9}$ for (c,f). Lines and shades represent the mean and standard error respectively.

## D.7 Ablation of Target Position for Computing `DETAIL`.

As a rule of thumb, for each demonstration, we generally want to take the embedding of its last few tokens because of the causal nature of inference and because information generally flows toward the end of the sequence [64]. We compare two possible choices of target position: the column position (immediately before the label) and the label position. We experiment on the demonstration removal task with these two choices of embeddings. The results are shown in Fig. 14. Using embeddings of both positions achieves decent task performance as reflected by the clear distinction in accuracy between removing demonstrations with high/low `DETAIL` scores, demonstrating that our method is robust against the choice of token embeddings. In our experiments, we adopt the column position to isolate information about the label from the embedding.

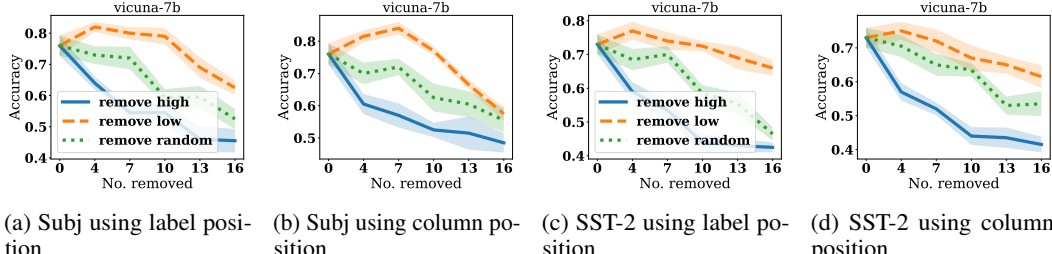

(a) Subj using label position

(b) Subj using column position

(c) SST-2 using label position

(d) SST-2 using column position

Figure 14: Results of model prediction accuracy vs. number of demonstrations removed using different positions for taking embeddings. $\lambda = 1.0$. Lines and shades represent the mean and standard error respectively.

## D.8 Experiment on State-Space Model Architecture

We consider experiments on a popular state-space model (SSM) architecture, a Mamba-2.8b model [27], which consists of 64 layers with $d = 2560$. The tasks are described in the main text in detail. While we find that DETAIL can still successfully attribute Mamba on certain tasks and datasets, the performance is inconsistent. One hypothesis is that the largest currently available Mamba model (2.8B) is still significantly smaller than the 7B LLMs we conduct experiments on in the main text. A smaller model size reduces the inductive power of the model to formulate the "internal optimizer", leading to less interpretive DETAIL scores. We would also like to note that DETAIL is not designed to work on SSMs.

**Demonstration removal.** The demonstration removal experiment follows the same setup as Sec. 5.2 but uses a Mamba-2.8b model instead, which is the largest model officially open-sourced. The results are shown in Fig. 15. Interestingly, removing demonstrations according to DETAIL scores still can influence predictive performance in the desirable manner where removing demonstrations with high $\mathcal{I}_{\text{test}}$ leads to lower accuracy and *vice versa*.

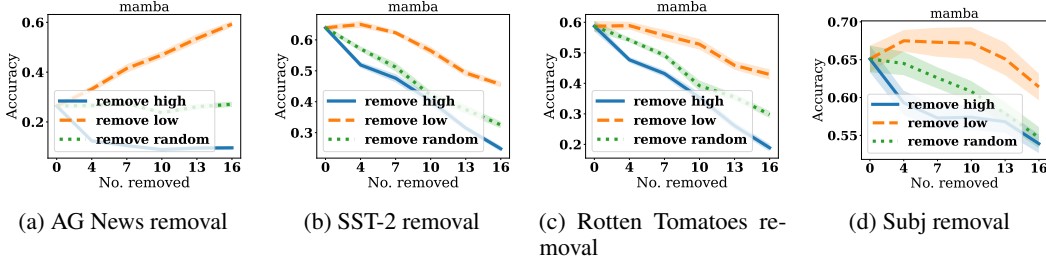

(a) AG News removal

(b) SST-2 removal

(c) Rotten Tomatoes removal

(d) Subj removal

Figure 15: Results of model prediction accuracy vs. number of demonstrations removed on Mamba. $\lambda = 1.0$. Lines and shades represent the mean and standard error respectively.

**Noisy demonstration detection.** We further consider the noisy demonstration detection task as described in Sec. 5.2 on Mamba. Unfortunately, the performance is not consistent across datasets, as shown in Fig. 16: detecting demonstrations with high $\mathcal{I}_{\text{self}}$ performs close to random selection on SST-2 and Rotten Tomatoes datasets, although the inference speedup is still significant. We leave the analysis of these failure cases to future work.

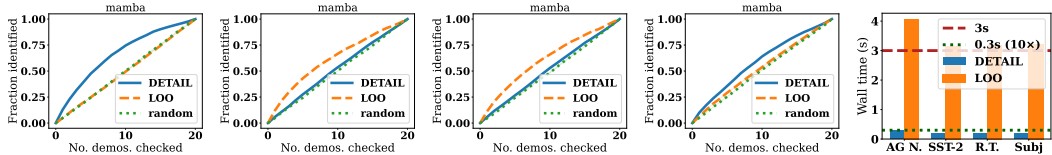

(a) Detecting noisy label on AG News

(b) Detecting noisy label on SST-2

(c) Detecting noisy label on Rotten Tomatoes

(d) Detecting noisy label on Subj

(e) Wall time comparison

Figure 16: (a-d) Fraction of all noisy labels identified vs. the number of demonstrations ranked by our method (with projection down to $1000$ dimension) and LOO checked respectively. (e) Wall time comparison across all datasets. $\lambda = 10^{-9}$. All experiments are repeated with $10$ independent trials. Lines and shades represent the mean and standard error respectively.

**Demonstration curation.** As DETAIL performs well using Mamba on the demonstration removal task, it is reasonable to hope that it works well on the demonstration curation task as well. As it turns out, DETAIL performs well on binary classification tasks as shown in Fig. 17 but performs poorly on AG News which is $4$-way classification. We hypothesize that this is due to Mamba's worse inductive power to formulate an internal algorithm successfully.

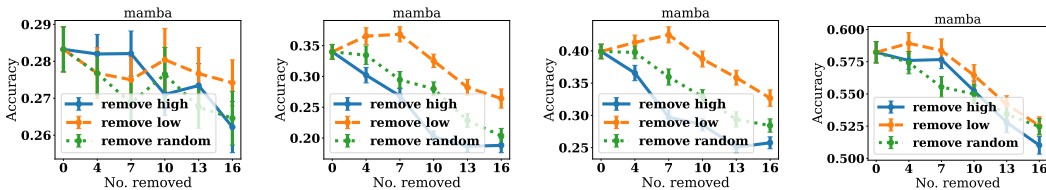

Figure 17: (Left to right) test accuracy vs. number of demonstrations removed using $\mathcal{I}_{\text{test}}$ on AG news, SST-2, Rotten Tomatoes, and Subj datasets using Mamba-2.8b. All experiments are repeated with $80$ independent trials. Lines and bars represent the mean and standard error respectively.

