# OpenReview forum: "DETAIL: Task DEmonsTration Attribution for Interpretable In-context Learning"
_NeurIPS.cc/2024/Conference — NeurIPS 2024 poster_

### Official Review · Reviewer_smfw · 2024-07-09

**Soundness:** 3
**Presentation:** 3
**Contribution:** 3
**Rating:** 7
**Confidence:** 4

**Summary:**

This paper introduces DETAIL, a novel technique for attributing and interpreting in-context learning (ICL) demonstrations in transformer-based language models. The authors propose an adaptation of the influence function, typically used in conventional machine learning, to address the unique characteristics of ICL. DETAIL treats transformers as implementing an internal kernelized ridge regression, allowing for efficient and effective attribution of demonstrations. The method is evaluated on various tasks, including demonstration perturbation, noisy demonstration detection, and real-world applications such as demonstration reordering and curation. The authors demonstrate DETAIL's superiority over existing attribution methods in terms of both performance and computational efficiency.

**Strengths:**

1. This paper proposes a novel approach, i.e., DETAIL to address the specific challenges of ICL attribution by leveraging the internal optimizer perspective of transformers.
2. The method incorporates random matrix projection to reduce dimensionality, resulting in significant speedups (up to 10x) while maintaining effectiveness.
3. DETAIL is shown to be effective across multiple tasks, including noisy demonstration detection, demonstration reordering, and curation, demonstrating its broad applicability.
4. The paper provides extensive experiments on both custom transformers and large language models (LLMs) like Vicuna-7b and Llama-2-13b, validating the method's effectiveness.
5. The authors demonstrate that DETAIL scores computed on white-box models can transfer to black-box models like GPT-3.5, enhancing its practical applicability.

**Weaknesses:**

1. Section 5.1 presents an evaluation on a custom transformer using the MNIST dataset. While this provides an initial demonstration of DETAIL's capabilities, the paper doesn't clearly justify why this evaluation is necessary given the subsequent experiments on large language models. It's not immediately apparent how insights from this simplified setting transfer to more complex LLMs, potentially making this section feel disconnected from the main contributions of the paper.
2. Figure 3 shows that even the Llama-2-13b model achieves only 60-70% accuracy on the AG News dataset without any perturbation. This is significantly lower than the typical performance range of 85-96% reported in the literature for this dataset. Could the authors give more insight about the performance of in context learning on the AG News dataset?

**Questions:**

see the weakness section

**Limitations:**

Yes, the authors adequately addressed the limitations

---

> ### Author Rebuttal · Authors · 2024-08-05
>
> # Response to the review of reviewer smfw
>
> Dear reviewer smfw,
>
> We would like to thank you for the review and for highlighting the novelty of our approach.
>
> We would like to address your concern as follows.
>
> ### 1. Connection between the MNIST experiment and the LLM experiment
> We wish to clarify that we provide the MNIST experiment mainly as a visualization to show how DETAIL works as we believe it is more intuitive to understand visual similarity than semantic similarity of text. We also wish to use the MNIST experiment to highlight that our approach applies to ICL with transformers in general although we focus on transformer-based language models. We appreciate your feedback on the disconnection between the two parts. We will make the transition smoother by highlighting the above clarification in our revision.
>
> ### 2. Accuracy on AG News
> We believe two main factors are causing the gap in test accuracy. First, we use only $10$ demonstrations for ICL while AG News typically requires more than $30$ demonstrations to reach SOTA performance (e.g., Figure 3 (c) of [1]). Second, we modify the label names in our experiments to make sure it does not carry the semantic meaning to enforce ICL behavior (lines 171-176). The original label names (Worlds/Sports/Business/SciTech) make it easier for the language model to understand the task, leading to higher accuracy. The performance with altered label can also be observed in Figure 3 (c) of [1] (plotted line with legend $\tau_{\text{RA}}-ICL$) and is similar to the accuracy reported in our work.
>
> Once again, we would like to thank you for taking the time to review. We hope that our response sufficiently addresses your concerns.
>
>
> [1] Jiaoda Li et. al., "What Do Language Models Learn in Context? The Structured Task Hypothesis", ACL 2024.

---

### Official Review · Reviewer_oPDt · 2024-07-12

**Soundness:** 3
**Presentation:** 2
**Contribution:** 3
**Rating:** 5
**Confidence:** 4

**Summary:**

This paper proposes a new method to estimate the influence of ICL examples to a query. This influence estimation can better help ICL learning, for example, reorder ICL examples and curating ICL examples.

**Strengths:**

- The motivation is clear, and the paper overall is well-written.
- The idea to estimate the influence of ICL examples is interesting, and can help better ICL learning for LMs.
- Results in Table 2 seem to show that DETAIL performs better than existing influence functions (token-level) and also some recent work that estimates ICL influences.

**Weaknesses:**

- More comprehensive experiments: for applying DETAIL, the authors only showed a few tasks (all classification) and on one major public model (Vicuna-7b). To show the effectiveness of DETAIL, more comprehensive experiment results should be provided: e.g., Table 1 and 3 should use more tasks other than classification, and more models (can the authors add results on Llama-2-13b?).

- DETAIL for ICL order optimization: based on the description in Section 5.3, it seems like the authors re-ordered the examples not just based on I_{self}, but also based on the trend plot in Figure 5. Does test accuracy w.r.t. perturbed position alway need to be computed first before deciding on the orders? This seems really expensive and also the choice of putting "two demonstrations" in the front seems very arbitrary. How would one use such a method for a random new task?

- The need to use white-box models and poor transfer results: DETAIL requires white-box model access, and the transfer result in Table 3 is not significant on the more realistic setting (no corruption). Also, can the authors show the transfer results with more models, e.g., LLama-2 to GPT-3.5, and Vicuna-7b to GPT-4?

**Questions:**

See above.

**Limitations:**

Discussed in Section 6.

---

> ### Author Rebuttal · Authors · 2024-08-06
>
> # Response to the review of reviewer oPDt
>
> Dear reviewer oPDt,
>
> Thank you for the review and for acknowledging the motivation and main idea of our work.
>
> We would like to address your concerns in the following. Kindly note that all citation numbers follow the reference list from the main paper.
> ### 1. Comprehensiveness of experiments
> We wish to first highlight that our experiments serve mainly to demonstrate the interpretability of DETAIL's attribution and the applicability of DETAIL in various scenarios including demonstration curation and re-ordering. For models, in addition to Vicuna-7b, we have also considered Llama-2-7b, Llama-2-13b models in the paper, as well as an SSM architecture model, Mamba-2.8B, with some of the results in Appendix.
>
> We acknowledge that while DETAIL itself can be useful in many tasks, our experiments only considers classification tasks currently. To extend our work to support other types of tasks, we may modify the loss function accordingly. For example, to support generation tasks where the ground truth is a sequence of desired outputs, we may modify the loss in Eq(3) to be an autoregressive cross-entropy loss instead of an MSE loss.
>
> We thank you for suggesting including experiments for Table 1 and Table 3 using Llama-2-13b. Due to the character limit, we include the additional experimental results together with the transfer results in the PDF in the global rebuttal. We provide an analysis of the additional results in bulletpoint 4 below.
>
> ### 2. Placement of demonstrations in the re-ordering task
> We would like to clarify that figure 5 serves to verify how test accuracy changes with the position of the corrupted demonstration. Figure 5 shows that placing corrupted demonstrations at the two ends generally leads to better test accuracy for various datasets (e.g., subj, SST-2, and Rotten tomatoes). Hence, we use the DETAIL score as a proxy for the corruptedness of the demonstration and place a few demonstrations with the highest DETAIL score in front. We expect the same trend to hold on a random new task and we can use the same technique (placing a few demonstrations with the highest DETAIL score in front). The additional experiments in bulletpoint 4 are all conducted by placing 2 demonstrations with the highest score in front.
>
> ### 3. White-box access and Transfer performance on re-ordering task
> We acknowledge that DETAIL specifically considers the white-box setting to provide interpretable attribution. While methods under a black-box setting (e.g., [41, 46]) may offer wider applicability, they may also have limited interpretability and poorer performance due to the absence of access to the model's internal working (as shown in Table 2). We highlight that **interpretable attribution of ICL demonstrations is the main focus and contribution of this work**.
>
> We acknowledge that transfer performance is less significant for the re-ordering task in Table 3 when no demonstration is corrupted. We think it is because GPT-3.5/GPT-4o can already achieve very high test accuracy with the baseline ordering ( baseline accuracy with no corrupted demo in Table 3 is on average 12.0 percentage points higher than that in Table 1).
>
> However, while reordering demonstrations may not improve performance when there are no corrupted demonstrations, it does not deteriorate performance either, nor does it require heavy extra computation (only 1 pass). As such, reordering offers a way to obtain **"free" potential performance gain without the costly curation or obtaining more demonstrations**.
>
> On your comment that it is less "realistic" to have corrupted demonstrations in the ICL dataset, we argue that although a carefully curated ICL dataset contains few corrupted demonstrations, in various real-world scenarios, ICL demonstrations might not come from a well-prepared dataset but are generated in real-time. For example, LLM-powered search engines (e.g., Perplexity.AI) obtain ICL demonstrations in real-time from web-scrapped data based on the user query, which very often contains corrupted demonstrations. Recent works that consider self-generating ICL demonstrations from LLM (e.g., Chen et. al. "SELF-ICL: Zero-Shot In-Context Learning with Self-Generated Demonstrations", EMNLP 2023) are also subjected to corrupted demonstrations due to LLM's hallucination. More importantly, as ICL demonstrations are typically generated in real-time in these scenarios, costly curation is not tractable and thus the proposed DETAIL-based re-ordering technique that can indeed be computed in real-time, can be highly effective.
>
> ### 4. Analysis of additional results on re-ordering task
>
> We kindly refer you to the PDF attached in the global author rebuttal for the additional experimental results.
>
> For Llama-2-13b, performance improvement with no corrupted demonstration is less significant compared to Vicuna-7b, although with 3 corrupted demonstrations, we can still observe 1-2% improvement. We attribute this to the stronger baseline performance (baseline (no corrupted) average accuracy on Llama-2-13b is 16.4 percentage points higher than that on Vicuna-7b).
>
> For transfer performance, we observe that the performance gain is not significant with no corrupted demonstrations. With 3 corrupted demonstrations, GPT-4o shows limited performance gain, which is probably because GPT-4o has stronger pretrained prior and can achieve very high test accuracy even with some bad demonstrations. However, for GPT-3.5, the performance improves by more than 3% in this scenario. On the other hand, when 6 demonstrations are corrupted, re-ordering shows a significant improvement of over 3% on GPT-4o as well. This demonstrates the benefit of re-ordering, even for large models, with noisy demonstration data.
>
>
>
> Thank you for providing a constructive review and the questions. We hope our response has helped address your concerns. We are glad to provide further clarifications.

---

> > ### Comment · Reviewer_oPDt · 2024-08-12
> >
> > Thanks the authors for the rebuttal and adding the new results. Given all the results so far, the overall trend becomes a bit more clear:
> > (1) the method gives more significant improvement under the corruption setting (which is less realistic, and given the authors' argument on similarity to search retrieval, it would be more convincing to provide such results, even with a simple RAG setup);
> > (2) the methods does not yield significant gains when the baseline performance is already high (for GPT-4o it's ok, but for GPT-3.5 the gains still vary quite a bit, like there's no gain on "Subj" under the no corruption setting). It would be more convincing if the authors can show more consistent results across models/settings, and provide results on more challenging tasks where the baseline performance is not good enough, and see if DETAIL can still yield gains.
> >
> > I will maintain my score given the limited effectiveness of the proposed method, but I won't object to accepting this paper.

---

> > > ### Author Response · Authors · 2024-08-13
> > > **Thank you for acknowledging our response**
> > >
> > > Thank you very much for acknowledging our response and suggesting a new experiment setting with RAG for search retrieval. We will include your suggestions, along with our clarifications and additional experimental results, in the revision of our work.

---

### Official Review · Reviewer_EuFx · 2024-07-12

**Soundness:** 3
**Presentation:** 3
**Contribution:** 3
**Rating:** 6
**Confidence:** 4

**Summary:**

The paper introduces DETAIL, a novel influence-function based attribution technique to estimate the influence of each example in the demonstration sequence for the given target query for in-context learning (ICL). The authors empirically validate DETAIL’s effectiveness in stylized experiments and LLMs. Additionally, they demonstrate DETAIL’s applicability in real-world tasks like demonstration reordering and curation and the transferability of DETAIL’s attribution scores from white-box to black-box models.

**Strengths:**

1. Innovative Application of Influence Functions: The application of influence functions to interpret in-context learning (ICL) is both innovative and intriguing, offering a fresh perspective on model interpretability.
2. Transferability to Black-Box Models: The method demonstrates promising results in transferring attribution scores from white-box models to black-box models, which significantly enhances its practical applicability.
3. Promising Performance: The empirical performance of DETAIL is promising, showcasing its potential effectiveness in real-world scenarios

**Weaknesses:**

1. Strong Assumptions: The work is built on the assumption that transformers implement an internal optimizer. This assumption, while supported by some theoretical proof in stylized settings, may not universally hold. The proof further assumes a learn-to-learn setting for ICL, which differs from the definition provided in lines 88-105 in this paper. Additionally, Equation 3 assumes the loss function of ICL is a ridge regression function without robust theoretical guarantees.

2. Lack of Analysis for Equation 6: Given the strong assumptions underlying the derived Equation 6, its reliability may be questionable.

3. Sensitivity to Demonstration Ordering: Recent work has demonstrated that ICL is not sensitive to ordering with more powerful demonstration retrievers and advanced language models, contradicted  with the conclusion in  line 35.

4. Limited Experimental Comparisons: The experimental validation of DETAIL in ICL demonstration curation could be strengthened. The authors should compare their method with other learning-free demonstration selection methods, such as those based on BERT embeddings.

5. Computational Cost: The computational cost of DETAIL is relatively high for demonstration curation or selection. Existing methods that utilize BERT embeddings are more computationally efficient.

**Questions:**

1. How do you obtain $y_{test}$ for $I_{test}$?
2. Have you explored initializing $m(x)$ and $y$ with BERT embeddings? This could be a potential area for improvement, leveraging pre-trained embeddings for better initialization.

**Limitations:**

Yes

---

> ### Author Rebuttal · Authors · 2024-08-06
>
> # Response to the review of reviewer EuFx
>
> Dear reviewer EuFx,
>
> Thank you for your review, compliment for the innovative application of the influence function in our work, and the acknowledgment of the real-world applications of DETAIL.
>
> We would like to address your concerns as follows. Kindly note that all citation numbers follow the reference list of the main paper.
>
> ### 1. Strong assumption
> We wish to clarify that while the pioneering work [51] provided the theoretical proof under the learn-to-learn setting, follow-up works have theoretically shown that transformers can implement internal optimizers **even when they are pre-trained on random instances** [2, 52]. [20] further demonstrated the internal optimizer phenomenon w.r.t. GPT.
>
> While we acknowledge that the kernelized ridge regression does not have a formal theoretical guarantee, it should be noted that the formulation closely follows that mentioned in [51] with slight modification, as we apply it with the influence function (lines 128-131). Following the kernelized ridge regression, equation 6 is then rigorously derived by following the exact definition of the influence function.
>
>
> ### 2. Sensitivity to demonstration ordering
> We would appreciate it if you could kindly provide sources that have made this claim so that we could better address this concern.
>
> Assuming that it is the case where very large models are indeed insensitive to the ordering of demonstrations, we highlight that the claim in line 35 is based on previous works considering relatively smaller language models (typically less than 10b) [35, 37].
>
> Smaller language models are also widely used in real-life scenarios because of their cost-effectiveness (a 7b model can be run on a personal PC). In these applications, the models typically suffer from the problem of being sensitive to demonstration ordering.
>
> Additionally, while it is understandable that very large models are more robust to re-ordering, recent work (e.g., Table 12 of Wu et. al., Arxiv 2405.16122) has shown that there can be up to 10% accuracy difference when large noise is injected by carefully optimizing the order of demonstrations on GPT-3.5.
>
> ### 3. Curation experiment with BERT embedding
> Thank you for suggesting the additional comparison baseline (i.e., curating demonstration with BERT-based methods). We have conducted an additional experiment under the setting of Table 2 using BERT score from a popular sentence-transformer model `sentence-transformers/msmarco-bert-base-dot-v5`. The results are tabulated below:
>
>
> | Metric | DETAIL ($d=1000$) | DETAIL ($d=50$) | BERT Score |
> | --- | --- | --- | --- |
> | Subj (Accuracy) | **0.747 (2.60e-02)** | 0.713 (2.43e-02) | 0.671 (2.43e-02) |
> | SST2 (Accuracy) | **0.607 (2.12e-02)** | 0.530 (4.09e-02) | 0.477 (2.54e-02) |
> | Rotten tomatoes (Accuracy) | 0.555 (1.94e-02) | **0.557 (2.32)** | 0.435 (1.33e-02) |
> | AG News (Accuracy) | **0.412 (1.35e-02)** | 0.387 (1.32e-02) | 0.355 (1.58e-02) |
> | |
> | Subj (Wall time) | 5.22 (1.17e-01) | 3.18 (4.63e-02) | **2.97 (3.05e-02)** |
> | SST2 (Wall time) | 4.88 (1.35e-01) | **2.65 (4.09e-02)** | 2.91 (2.78e-02) |
> | Rotten tomatoes (Wall time) | 5.11 (1.06e-01) | 2.97 (5.36e-02) | **2.95 (4.43e-02)** |
> | AG News (Wall time) | 10.4 (1.07e-01) | 6.17 (7.30e-02) | **3.15 (5.87e-02)** |
>
> While we acknowledge that BERT score is relatively faster than DETAIL ($d=1000$), the speed gain is mainly due to **BERT scores being model-agnostic** and thus likely comes at the cost of poorer performance compared to DETAIL (as reflected by the lower accuracy in the table above). We wish to highlight that one advantage of **DETAIL is that its attribution is dependent on the transformer used** which provides more interpretability and improved performance.
>
> Moreover, while BERT score is fast, we note that DETAIL is also very computationally efficient: DETAIL is much faster than most attribution methods shown in Table 2. DETAIL can be even faster by reducing $d$. As shown in the table, when $d=50$, DETAIL has a comparable running time as BERT score while still achieving higher accuracy.
>
>
> ### Questions
> 1. Obtaining $y_{\text{test}}$: We wish to clarify that $y_{\text{test}}$ is part of $z_{\text{test}} = (x_{\text{test}}, y_{\text{test}})$ (line 103) which refers to the ground truth label of the query sample. $y_{\text{test}}$ is provided as part of the test dataset.
>
> 2. Incorporating BERT embedding in $m(x)$: Thank you for providing the insightful suggestion to incorporate BERT embedding in calculating $m(x)$. We believe it is an exciting direction for future research to consider a combination of BERT and transformer embeddings to improve performance. We have conducted a preliminary experiment with three different $m(x)$ computation tabulated below. For the transformer embedding, we use $d=768$, the same size as the BERT embedding. When using BERT embedding only, the performance is worse than DETAIL. Interestingly, using an equal-weighted average results in SOTA test accuracy on 3 out of 4 datasets.
>
> | Metric | Bert Embedding Only | Weighted Average (equal weight) | Concatenation |
> | --- | --- | --- | --- |
> | Subj (Accuracy) | 0.665 (2.36e-02) | **0.758 (1.97e-02)**  | 0.719 (2.75e-02) |
> | SST2 (Accuracy) | 0.475 (1.31e-02) | 0.579 (2.34e-02)  | 0.596 (1.96e-02) |
> | Rotten tomatoes (Accuracy) | 0.510 (1.85e-02) | **0.575 (2.46e-02)**  | 0.570 (2.75e-02) |
> | AG News (Accuracy) | 0.408 (2.10e-02) | **0.425 (1.76e-02)**  | 0.412 (1.50e-02) |
> | |
> | Subj (Wall time) | 5.96 (3.05e-01) | 8.05 (1.73e-01)  | 7.89 (1.33e-01) |
> | SST2 (Wall time) | 5.24 (1.63e-01) | 7.61 (1.69e-01)  | 7.51 (2.01e-01) |
> | Rotten tomatoes (Wall time) | 5.25 (2.93e-01) | 9.18 (1.78e-01)  | 9.11 (1.57e-01) |
> | AG News (Wall time) | 4.30 (8.07e-02) | 12.8 (2.00e-01)  | 11.2 (1.73e-01) |
>
>
> Thank you again for providing a constructive review and offering insightful suggestions. Please let us know if there are any remaining questions. We are glad to offer further clarifications.

---

> > ### Comment · Reviewer_EuFx · 2024-08-12
> > **Reply to Authors' Rebuttal**
> >
> > Thanks for the authors' detailed responses to my questions. The additional experiments and discussions are helpful and provide more evidence supporting its usability. Hence, I would like to raise my score.

---

> > > ### Author Response · Authors · 2024-08-13
> > > **Thank you for acknowledging our response and raising the score**
> > >
> > > Thank you very much for acknowledging that the additional experiments in our response are detailed and helpful and for raising the score. We will incorporate the experimental results and our clarifications in our revision.

---

### Official Review · Reviewer_ixwr · 2024-07-13

**Soundness:** 4
**Presentation:** 3
**Contribution:** 3
**Rating:** 8
**Confidence:** 4

**Summary:**

The paper proposes a novel attribution method for demonstrations in ICL. The proposed method takes a perspective that the transformers learn in context by formulating an internal optimizer. The influence function is approximated as an internal kernelized ridge regression, where the representations are taken from the intermediate layers of the transformers and white-box LLMs. The paper also demonstrate that the attribution obtained from the white-box LLM exhibits transferable characteristics to black-box models. The paper also showcases a few promising applications of DETAIL.

**Strengths:**

- By and large, the paper is written well. I especially appreciated the discussion on the relations to reinforcement learning, and how the potential functions differ from Lyapunov functions used in more classical settings.
- The idea of using the perspective that transformers formulating an internal optimize for attribution is novel and useful.
- The section on accelerating the computation with self-influence is also interesting, and highlights the authors attention to computational issues.
- The application of the DETAIL may open new research directions in LLMs for bettter instruction-tuning algorithms.
- The mathematical derivations appear to be correct.

**Weaknesses:**

- It would have been interesting to see how well this method would identify adversarial attacks for LM.
- Some additional commentary on how DETAIL can help with LLM training would have been very helpful.

**Questions:**

- Could you comment on whether DETAIL can be used to discover adversarial attacks of LLMs?
- Could you comment on how DETAIL can help with LLM training especially for instruction tuning?

**Limitations:**

Yes, the author addressed the limitations.

---

> ### Author Rebuttal · Authors · 2024-08-06
>
> # Response to the review of reviewer ixwr
>
> Dear Reviewer ixwr,
>
> Thank you for reviewing our paper and highlighting the novelty of our application of 'influence' for ICL which has the potential to inspire better instruction-tuning algorithms.
>
> We would like to address your concerns in the following.
>
> ### 1. Identifying adversarial attacks for LM
> We thank you for suggesting this exciting potential use case. In this work, we have demonstrated how DETAIL can identify corrupted demonstrations, which is one form of adversarial attack where an attack is performed by perturbing the ICL demonstration label (Table 1 and Table 3). We believe it is an interesting area to research how DETAIL can be applied to other types of attacks. For example, DETAIL might be used to defend against attacks that attempt to steal private ICL demonstrations by filtering out or rephrasing queries that cause excessively high DETAIL scores.
>
> ### 2. Additional comment on how DETAIL can help with LLM training
> We thank you for the question. While we primarily focus on in-context learning settings in this work, the DETAIL score can also potentially be used for selecting demonstrations for instruction fine-tuning. One way to perform the selection is to compute the DETAIL score of each ICL demonstration on a validation set and then use the demonstrations with the highest DETAIL scores for fine-tuning.
>
> We will incorporate the mentioned points in the future work section in our work.
>
> Once again, we would like to thank you for the insightful suggestions. We hope our response has been helpful in addressing your questions.

---

### Author Rebuttal · Authors · 2024-08-06

We thank all the reviewers for taking the time to provide constructive reviews and insightful suggestions. We have addressed the concerns in the respective rebuttal sections. As a general response, we would like to highlight that the main contribution of this work is to provide an interpretable attribution for ICL demonstrations on transformers that is also computationally efficient. Our experiments demonstrate DETAIL's effectiveness in attribution as well as in applications such as curation and re-ordering.

We have attached a PDF below including additional experiments on the re-ordering task. For additional experiments that consider BERT-related methods, please refer to our response to reviewer EuFx.

We are more than happy to clarify any additional questions regarding our rebuttal during the author-reviewer discussion period.

---

### Comment · Area_Chair_1JTS · 2024-08-12

Dear reviewers: as you are aware, the reviewer-author discussions phase ends on Aug 13. We request you to kindly make use of the remaining time to contribute productively to these discussions. If you have not read and/or responded to author rebuttal, please do it asap so that the authors get a chance to respond to you. If you have more questions to ask or want further clarification from the authors, please feel free to do it.

---

### Decision · Program_Chairs · 2024-09-25

**Decision:**

Accept (poster)

**Comment:**

The paper uses influence function-based method for in-context learning. All reviewers agree that the submission can be accepted. The method is simple yet effective. The results are positive and can be transferred from white-box models to other black-box models.